# Mechanistic insights into glycoside 3-oxidases involved in *C*-glycoside metabolism in soil microorganisms

André Taborda [1,5], Tomás Frazão [1,5], Miguel V. Rodrigues[1],
Xavier Fernández-Luengo[2], Ferran Sancho [3], Maria Fátima Lucas [3],
Carlos Frazão[1], Eduardo P. Melo [4], M. Rita Ventura [1], Laura Masgrau [2,3],
Patrícia T. Borges [1] & Lígia O. Martins [1] ✉

*C*-glycosides are natural products with important biological activities but are recalcitrant to degradation. Glycoside 3-oxidases (G3Oxs) are recently identified bacterial flavo-oxidases from the glucose-methanol-coline (GMC) superfamily that catalyze the oxidation of *C*-glycosides with the concomitant reduction of $O_2$ to $H_2O_2$. This oxidation is followed by C-C acid/base-assisted bond cleavage in two-step *C*-deglycosylation pathways. Soil and gut microorganisms have different oxidative enzymes, but the details of their catalytic mechanisms are largely unknown. Here, we report that *Ps*G3Ox oxidizes at 50,000-fold higher specificity ($k_{cat}/K_m$) the glucose moiety of mangiferin to 3-keto-mangiferin than free D-glucose to 2-keto-glucose. Analysis of *Ps*G3Ox X-ray crystal structures and *Ps*G3Ox in complex with glucose and mangiferin, combined with mutagenesis and molecular dynamics simulations, reveal distinctive features in the topology surrounding the active site that favor catalytically competent conformational states suitable for recognition, stabilization, and oxidation of the glucose moiety of mangiferin. Furthermore, their distinction to pyranose 2-oxidases (P2Oxs) involved in wood decay and recycling is discussed from an evolutionary, structural, and functional viewpoint.

*C*-glycosides represent a large group of natural products in which the anomeric carbon of glucose is directly connected via carbon-carbon bonding to an aglycone moiety (anthraquinone, flavone, terpenoids, phenols, among others). These compounds are secondary metabolites produced by plants and microorganisms and exhibit great structural diversity, wide natural distribution, and significant biological activities, including antioxidant, anti-inflammatory, antibacterial, antiviral, and antitumor[1,2]. *C*-glycosides have shown increasing importance in the pharmaceutical, agricultural, and food industries, and a great effort

has been focused on their synthesis[3,4]. Several *C*-glycosides like puerarin (daidzein 8-C-β-D-glucoside) have been the precursor of clinical drugs, and biotechnological strategies have been optimized for large-scale production of plant *C*-glycosides via heterologous expression systems[2,5].

Gut microbiota that performs bioconversion of *C*-glycosides to aglycones with beneficial health effects has been identified over the past decade[6–8]. However, very recently, microbial screenings and biochemical studies suggest a ubiquitous process of C−C bond cleavage

[1]Instituto de Tecnologia Química e Biológica António Xavier, Universidade Nova de Lisboa, Av da República, 2780-157 Oeiras, Portugal. [2]Department of Chemistry, Universitat Autònoma de Barcelona, 08193 Bellaterra, Spain. [3]Zymvol Biomodeling, C/ Pau Claris, 94, 3B, 08010 Barcelona, Spain. [4]Centro de Ciências do Mar, Universidade do Algarve, 8005-139 Faro, Portugal. [5]These authors contributed equally: André Taborda, Tomás Frazão. ✉e-mail: lmartins@itqb.unl.pt

reactions in nature[9,10]. Compared with other glycosides (*O*-, *N*- and *S*-glycosides), *C*-glycosides are considerably more stable against chemical and enzymatic treatments. Because of this, *C*-glycosides are not deglycosylated by glycoside hydrolases, the so-called glycosidases. Instead, their microbial catabolic pathway includes enzymes that catalyze an oxidative step followed by the C–C bond cleavage[9]. Soil and intestinal microorganisms have similar C–C bond-cleaving enzymes[9]. In contrast, the initial oxidation step by *C*-glycoside-3-oxidases that oxidize the C3 of the sugar moiety, is catalyzed by NAD(H) anaerobic oxidoreductase in intestinal microorganisms[6,7,11] and by FAD-dependent oxidoreductases in soil microorganisms[8,10]. These latter enzymes display ~60% similarity to pyranose oxidases (POx, pyranose: oxygen 2-oxidoreductase; EC 1.1.3.10) from the glucose-methanol-choline (GMC) superfamily of enzymes[12].

G3Oxs from soil bacteria such as *Microbacterium* sp. 5-2b CarA, *Arthrobacter globiformis* NBRC12137 *Ag*CarA, and *Microbacterium trichothecenolyticum* NBRC15077 *Mt*CarA do not exhibit detectable activity for glucose and instead, oxidize *C*-glycosides such as carminic acid, mangiferin, and C6-glycosylated flavonoids, as well as *O*-glycosides but at a significantly lower rate, at C3 to form the corresponding 3-keto glycosides[10]. An analogous substrate preference was observed in bacterial *Sc*P2Ox from *Streptomyces canus* that displayed 100 to 1000-fold higher enzymatic activity towards the oxidation of *C*-glycoside puerarin compared to monosaccharide[13]. It was suggested that FAD-dependent G3Oxs from the POx family are ancestors of enzymes that oxidize glucose at the C2 position, the pyranose 2-oxidases (P2Oxs)[10,13]. The fungal P2Oxs, have most likely been acquired by horizontal gene transfer from bacteria and could have evolved and specialized over time to oxidize lignocellulose-derived sugars such as D-glucose, D-xylose, or D-galactose. Fungal P2Oxs are secreted to the extracellular space and are involved in wood decay and recycling. They are the most extensively studied POx, particularly from *Trametes multicolor*, *Peniophora* sp, and *Phanerochaete chrysosporium*[14–16]. They comprise a highly conserved flavin-binding domain with a Rossman-like-fold where FAD covalently binds and a substrate-binding domain. P2Oxs are homotetrameric, and the access to the active site is restricted by four channels that route the substrate from the enzyme surface to the active site cavity[17–19]. The structural characterization of bacterial G3Ox, *Mt*CarA, and *Sc*P2Ox revealed a few structural and functional aspects of these enzymes[10,13]. However, many fundamental questions remain, particularly the mechanisms behind the diverse substrate specificities.

Here, we investigated the bacterial enzyme *As*P2Ox from *Arthrobacter siccitolerans* (now *Pseudarthrobacter siccitolerans*[20,21]), using a combination of experimental and computational approaches that reveal functional and structural details that explain these enzymes ability to bind and oxidize larger glycoside substrates. This work contributed to unveiling the catalytic mechanism of a critical catabolic enzyme involved in the degradation of recalcitrant *C*-glycosides in nature that remain to be fully disclosed and advancing our understanding of the structure-function relationships among members of the POxs family of enzymes.

## Results and discussion
### Biochemical and kinetic characterization
The pseudo-second-order constants measured by transient state kinetics for the reductive-half reaction using D-Glc as electron donor and for oxidative-half reactions ($k_{red}^{Glc} = 0.15 \pm 0.02\,M^{-1}\,s^{-1}$ and $k_{ox} = (0.76 \pm 0.06) \times 10^6\,M^{-1}\,s^{-1}$) indicate that the rate-limiting step of *Ps*P2Ox is the reductive-half reaction, i.e., the oxidation of D-Glc to 2-keto-D-Glc, similarly to other studied POx (Fig. 1a and Supplementary Fig. 1)[22,23]. The compound used to measure *Ps*P2Ox enzymatic activity in the coupled assay with horseradish peroxidase (HRP)[20], 2,2′-azino-bis(3-ethylbenzothiazoline-6-sulfonic acid) (ABTS) is after oxidation to a green cation radical, reduced by the enzyme to the dicationic form,

leading to an underestimation of the enzymatic activity (Supplementary Table 1). Therefore we have optimized the assay using substrates 4-aminophenazone (AAP) and 3,5-dichloro-2-hydroxybenzene-sulfonic acid (DCHBS), which originate after oxidation a pink chromogen, N-(4-antipyryl)-3-chloro-5-sulfonate-p-benzoquinone-monoimine[24], which is not reduced by *Ps*P2Ox. This assay allowed re-estimate the catalytic parameters for monosaccharides D-Glc, D-Xyl, D-Gal, D-Ara, and D-Rib (Table 1 and Supplementary Fig. 2). Still, the obtained results confirmed that the enzyme is a poor biocatalyst for the oxidation of monosaccharides. In contrast, and notably, the $K_m$ of *Ps*P2Ox for molecular oxygen is 1–3 orders of magnitude lower than the described for other POxs (Supplementary Table 2). Following the substrate preference of recently characterized CarA enzymes[10] and *Sc*P2Ox[13] towards the oxidation of *C*- and *O*-glycosides, the activity of *Ps*P2Ox was tested for mangiferin (Mang), carminic acid, and rutin (Supplementary Fig. 3); *Ps*P2Ox is inactive towards carminic acid and rutin; however, Mang is oxidized at a catalytic efficiency ($k_{cat}/K_m$) four orders of magnitude higher than D-Glc, displaying a $k_{cat}$ around 40-fold higher and a $K_m$ 1000-fold lower (Table 1). Additionally, transient state kinetics were also performed to estimate the pseudo-second-order constants of the reductive-half reaction using Mang as substrate, $k_{red}^{Mang} = (0.27 \pm 0.02) \times 10^5\,M^{-1}\,s^{-1}$, further supporting that Mang is a preferred substrate than D-Glc (Fig. 1a, Supplementary Fig. 1).To identify the product(s) of the reaction, enzymatic reactions with Mang were monitored by TLC (Supplementary Fig. 4) and characterized using a combination of 1D and 2D-NMR (Supplementary Figs. 5, 6, and 7 and Supplementary Table 3). The NMR data showed that oxidation must occur at the 3-OH, affording the 3-keto mangiferin (Fig. 1b). Similarly to Mang $^1$H NMR, the anomeric proton of the reaction product was easily identified at 5.17 ppm (Supplementary Fig. 5). Using this peak as a starting point, COSY NMR (Supplementary Fig. 7a) was used to assign neighbor protons: the anomeric H1 correlates to the H2 (4.69 ppm), and the H5 (3.27 ppm) is correlated to H4 (4.14 ppm) and H6 (3.55 ppm). The peaks of H2 and H4 are doublets (Supplementary Table 3 and Supplementary Fig. 7a), indicating the absence of the coupling constants with a vicinal H3. The chemical shift values of H2 and H4 also appear to have shifted downfield, as well as their carbons (correlated through HMQC in Supplementary Fig. 7b), indicating the existence of a more vicinal electronegative group due to the oxidation at C3. Due to the significantly higher specificity of the enzyme towards the C3 position of Mang compared to the oxidation of the C2 position of D-Glc, we renamed this enzyme from *Ps*P2Ox to *Ps*G3Ox. The phylogenetic analysis of bacterial and fungal POxs (Fig. 1c) corroborates a clear division between those with a higher specificity towards the oxidation of D-Glc and other monosaccharides at the C2 position, represented by well-characterized fungal P2Oxs (Fig. 1c, green clade), and those with lower specificity for D-Glc and higher specificity towards the oxidation of *C*-glycosides at the C3-position, the bacterial G3Ox (Fig. 1c, blue clade). Interestingly, the bacterial *Ka*P2Ox, the closest bacterial member to fungal counterparts, shows a high specificity for D-Glc[25] (Supplementary Table 2). To investigate the presumable role of *Ps*G3Ox in *C*-glycosides catabolism, a Basic Local Alignment Search Tool (BLAST) was performed in the genome of the drought-tolerant *P. siccitolerans* to find putative *C*-deglycosydases in the neighborhood of *Ps*G3Ox coding gene. The *Ps*G3Ox belongs to a gene cluster that has a similar organization to other soil bacteria[9]. Downstream of the *psg3ox* gene, two homologs, *Ps*CGD_C1, and *Ps*CGD_C2, were found showing 37% and 33% identity when compared to CarC enzyme and two other ORFs, *Ps*CGD_B1 and *Ps*CGD_B2, show 49% and 38% identity to CarB, which are *C*-glycoside deglycosydases (CGDs) that catalyze the cleaving of the C-C bond between the sugar and the aglycone moiety[9] (Fig. 1d). These results hint at the role of *Ps*G3Ox as part of the *C*-deglycosylation catabolic pathway with a putative biological function in, e.g., sugar uptake from natural glycosylated compounds that can act as carbon sources for the microorganism metabolism.

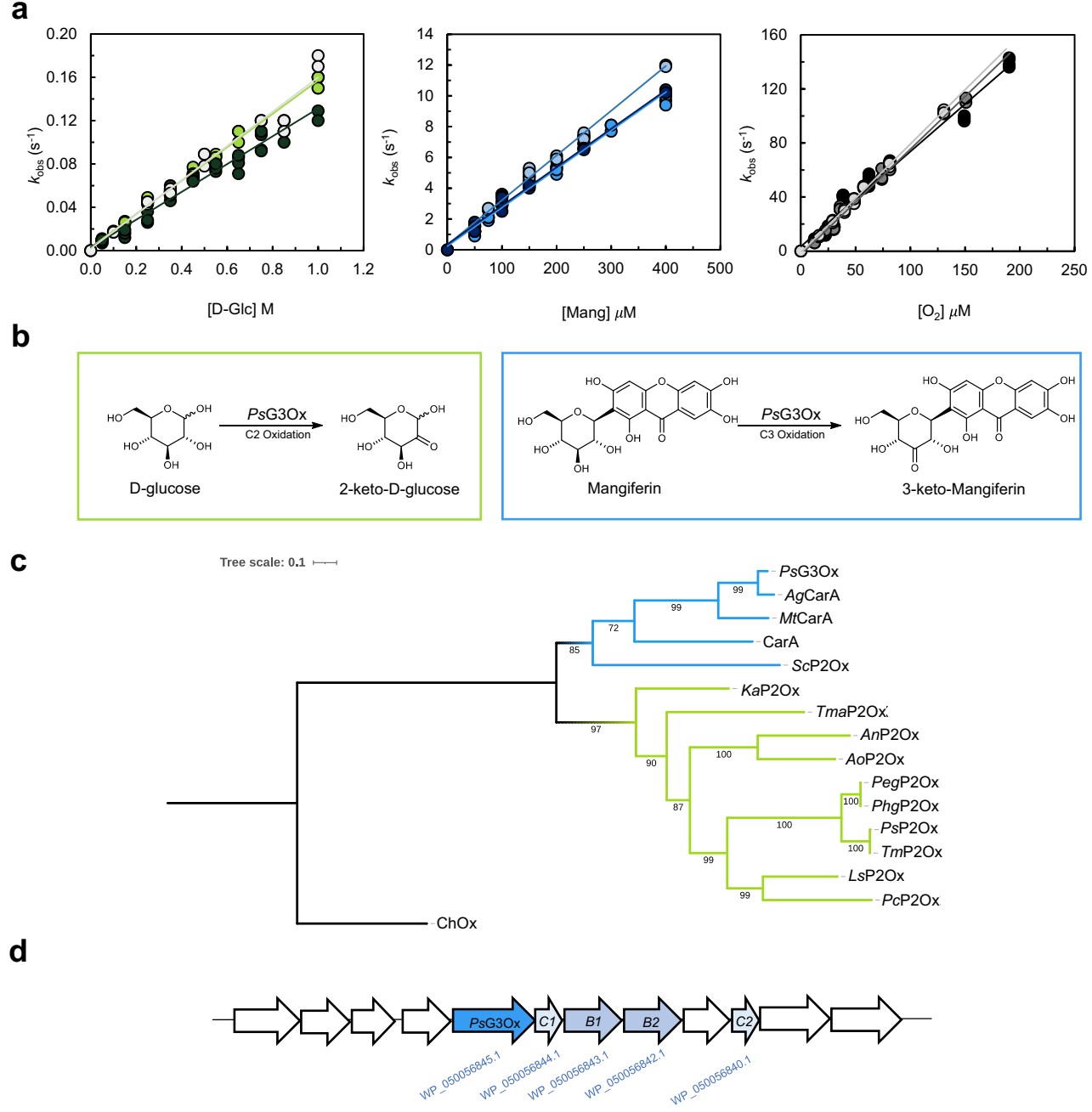

**Fig. 1 | Reaction scheme of *Ps*G3Ox, phylogenetic analysis of P2Oxs/G3Oxs and *Ps*G3Ox *C*-deglycosilation gene cluster. a** Transient-state kinetics for the reductive half-reaction using D-Glc (left) or Mang (middle) as electron donor and the oxidative half-reaction (right). All transient-state data were obtained using a stopped-flow apparatus under an anaerobic atmosphere at 25 °C. The routine buffer was 100 mM sodium phosphate buffer with 200 mM NaCl at pH 7.5. Independent experiments (*n* = 3) are evidenced in different color shades. Source data are provided as a Source Data file. Some traces for both half-reactions are shown in Supplementary Fig. 1. **b** The reaction catalyzed by *Ps*G3Ox to convert D-Glc to 2-keto-D-Glc[20] and the conversion of Mang to 3-keto-Mang. **c** Maximum likelihood phylogenetic relationship between characterized POxs; cholesterol oxidase sequence, ChOx, from *Streptomyces* sp. was used as outgroup. The blue cluster represent the bacterial enzymes that shows higher specificity for glycosides and the green cluster group the dimeric bacterial enzyme (*Ka*P2Ox) and the tetrameric fungal enzymes that display higher specificity towards D-Glc. The phylogenetic tree was generated using Molecular Evolutionary Genetics analysis (MEGA 11) software and the input sequences are provided in the Source Data file. **d** The gene coding for *Ps*G3Ox is part of a putative *C*-deglycosylation catabolic pathway. Four proteins with similarity to known *C*-glycoside deglycosydases (CGDs) are encoded in close vicinity to the *psg3ox* gene in *P. siccitolerans* 4J27 genome: *Ps*CGD_C1 and *Ps*CGD_C2 display 37%, and 33% identity compared to CarC and *Ps*CGD_B1 and *Ps*CGD_B2 which displayed 49% and 38% identity compared to CarB present from *Microbacterium* sp. 5-2b[9].

## PsG3Ox overall structure

The crystal structure of PsG3Ox was refined to 2.01 Å resolution, and the crystal structures of PsG3Ox-Glc and PsG3Ox-Mang complexes were refined at 2.35 and 2.60 Å resolutions (Supplementary Table 4). The enzyme crystallized as a monomer, and all crystals belong to space group C222₁, showed similar cell dimensions, and contained one molecule in the asymmetric unit (a.u.). The D-Glc and Mang were refined with substoichiometric occupancies (~0.8) (Supplementary Fig. 8) to have similar atomic displacement parameters (a.d.p.s) with neighboring atoms, 57–63 Å² vs. 56–70 Å² and 85–117 Å² vs. 67–95 Å², respectively. The weak electron density in the aglycone moiety of Mang suggests that this region is more flexible, presenting higher a.d.p.s values than the glucose moiety, 108–117 Å² vs. 85–96 Å². The lower number of interactions established between the aglycone and the enzyme could have contributed to the higher flexibility of this region. The overall structure of PsG3Ox is composed of a flavin- and a substrate-binding domain, comparable to other POx structures (Fig. 2a). PsG3Ox shows a root-mean-square deviation (r.m.s.d.) value of 0.84 Å compared with homologous Cα positions of the recently characterized bacterial MtCarA (PDB 7DVE)[10]. This is in contrast to the significantly higher r.m.s.d. values (1.70–1.81 Å) of structures of P2Oxs from the fungal origin, T. multicolor (PDB 1TT0), Peniophora sp. (PDB 1TZL) and P. chrysosporium (PDB 4MIF). PsG3Ox shows when compared to the fungal enzymes (Fig. 2b, c), (i) a significantly more solvent-exposed FAD cavity, (ii) smaller monomers' size (~500 residues instead of the ~600 residues), (iii) a monomeric structure (following the reported monomeric oligomerization state in solution[20]) instead of the tetrameric fungal state, and, (iv) significantly higher <a.d.p.>'s values (52 Å²) than fungal enzymes (12–30 Å²).

The access to the FAD is made through a cavity (Fig. 2d, Supplementary Table 5) that contains (i) residues ¹²⁵AAHW¹²⁸ that are at the same structural position of fungal flavinylation motif, ¹⁶⁵STHW¹⁶⁸ that covalently binds FAD in TmP2Ox, (ii) the substrate loop (³⁴⁶ASPVPLADD³⁵⁴), which in fungal enzymes, is reportedly a dynamic gating segment that fine-tunes the enzymes' reactivity[15,26], and (iii) the insertion-1 segment (between residues 60–93). Furthermore, the analysis of the PsG3Ox active site pocket unveiled the presence of a 25 Å length tunnel that connects the enzyme surface and the reactive isoalloxazine moiety of FAD (Fig. 2e). This tunnel is mostly delimited by hydrophobic residues (Supplementary Table 5) and has a radius ranging from 1.7 to 2.0 Å. The presence of hydrophobic tunnels allows an efficient way to deliver molecular oxygen to the enzyme's active site[27] and in PsG3Ox, this hydrophobic tunnel can provide a route for O₂ (with a van der Waals radius of 1.50–1.55 Å) to reach the FAD^{N5} where it is reduced to hydrogen peroxide. Due to the polarity of H₂O₂ its release should occur alternatively through the FAD cavity (Fig. 2f). Noble gases like krypton or xenon have been used in the investigation of hydrophobic tunnels due to their hydrophobicity and high atomic numbers[28–30]. The similar dimensions of krypton atoms, with O₂ minimal dimensions, make this a convenient gas to locate possible O₂ tunnels. Pressurization experiments with Kr inserted four krypton atoms in the PsG3Ox crystal structure; one of the Kr atoms (surrounded by hydrophobic residues V299, V349, L351, G366, and F368) is located in an internal pocket connected to the bottom section of the tunnel (Supplementary Fig. 9). This pocket is also present in the PsG3Ox non-pressurized structure. These results suggest that the hydrophobic tunnel connecting the FAD with the solvent can be used as a diffusion path for molecular oxygen in PsG3Ox.

The structural alignment revealed that PsG3Ox (as well as MtCarA) display nine deletions and four insertions when compared to fungal enzymes (Fig. 2h). Two of the deletions (boxes 1 and 3) correspond to the oligomerization loop and arm regions involved in the fungal P2Ox inter-subunit interactions (Fig. 2a)[17]. An additional deletion (boxes 9, 10, and 11) corresponds to a region in fungal enzymes known as the head domain. The major insertion in PsG3Ox is the insertion-1 segment

**Table 1 | Apparent steady-state kinetic parameters of wild-type and variants PsG3Ox for different substrates**

| Enzyme | Substrate | $k_{cat}$ (s⁻¹) | $K_m$ (M) | $k_{cat}/K_m$ (M⁻¹ s⁻¹) |
|---|---|---|---|---|
| Wild-type | D-Glucose | 0.19 ± 0.03 | 0.46 ± 0.13 | 0.45 ± 0.09 |
| Wild-type | D-Xylose | 0.13 ± 0.02 | 1.00 ± 0.20 | 0.13 ± 0.01 |
| Wild-type | D-Galactose | 0.02 ± 0.00 | 0.58 ± 0.16 | 0.03 ± 0.01 |
| Wild-type | L-Arabinose | 0.02 ± 0.00 | 0.57 ± 0.04 | 0.03 ± 0.00 |
| Wild-type | D-Ribose | 0.01 ± 0.00 | 0.19 ± 0.08 | 0.06 ± 0.01 |
| Wild-type | Dioxygen[a] | 0.14 ± 0.04 | (5.83 ± 2.10) × 10⁻⁶ | (19.87 ± 4.82) × 10³ |
| Wild-type | Mangiferin | 8.13 ± 1.67 | (0.49 ± 0.10) × 10⁻³ | (19.22 ± 2.71) × 10³ |
| Wild-type | Rutin | nd | – | – |
| Wild-type | Carminic acid | nd | – | – |
| Δloop (345-359) | D-Glucose | 0.06 ± 0.01 | 0.36 ± 0.13 | 0.19 ± 0.06 |
| Δloop (345-359) | Mangiferin | 1.4 ± 0.4 | (0.08 ± 0.02) ×10⁻³ | (23.6 ± 0.3) × 10³ |
| Δinsert1 (73-93)[b] | D-Glucose | – | – | (0.19 ± 0.01) × 10⁻² |
| Δinsert1 (73-93)[b] | Mangiferin | [c] | | |

The catalytic parameters for D-Glc and other monosaccharides were estimated using the HRP-AAP/DCHBS coupled assay; for molecular oxygen, the reactions were followed in an Oxygraph in the presence of 1 M D-Glc. The reactions with Mang were monitored by oxygen consumption in an Oxygraph. All reactions were performed in 100 mM sodium phosphate buffer at pH 7.5 and 37 °C. The data is represented as mean ± SD of the independent experiments (n = 3). The kinetic parameters were determined by fitting the data directly on the Michaelis-Menten equation using OriginLab. Source data are provided as a Source Data.
[a]D-Glc was used as an electron donor; nd – not detected.
[b]Assays performed after in vitro flavinylation of the purified preparation; $k_{cat}/K_m$ was obtained from the first-order approximation of the Michaelis–Menten equation ([S] ≪ $K_m$).
[c]Residual activity was detected using 2.5 mM of Mang ($V_{max}$ = 13.4 ± 1.7 nmol min⁻¹ mg⁻¹).

containing 33-residues (box 2), forming two α-helices, α2 (60–71) and α3 (83–88), together with two loops (72–82 and 89–93), which are close to the substrate-binding domain and in the neighboring of the FAD access (Fig. 2d and Supplementary Fig. 10). Interestingly, insertion-1 is located in an equivalent structural position to the oligomerization domains of fungal enzymes (Supplementary Fig. 11a), i.e., on the interaction interfaces; region 60–70 of PsG3Ox is close to the fungal oligomerization loop, while region 71–93 (the most flexible part of insertion-1) is located near the fungal oligomerization arm domain. Insertion-1 is also present in bacterial homologs (Supplementary Fig. 11b–f), except for the dimeric bacterial KaP2Ox, which contains oligomerization domains comparable to those in fungal enzymes (Supplementary Fig. 11g). The segments that differentiate monomers from oligomers tend to be located on the interaction interfaces, where they mediate or disrupt oligomerization and are usually loops[31]. It has been claimed that homooligomerization in glycosyltransferases and other proteins might be crucial for their function[32]. We speculate that the evolutionary mechanism of POxs homooligomerization can hypothetically occur through the deletion and insertion of segments in the region where insertion-1 is located, promoting the stabilization of dimers and tetramers[33].

## Catalytic center in PsG3Ox

The FAD cofactor in PsG3Ox is non-covalently bonded, similar to other characterized bacterial enzymes, except KaP2Ox, and in contrast to fungal P2Oxs where FAD is covalently linked through its 8α-methyl group, e.g., H167^{NE2} in TmP2Ox (PDB 2IGK). PsG3Ox's equivalent residue, H127, is approximately 6.5 Å away from the FAD^{C8M} atom (Fig. 2g). In vitro deflavinylation (Supplementary Fig. 12), followed by FAD incorporation, revealed that Apo-PsG3Ox binds exogenous FAD with an estimated dissociation constant $K_D$ = (2.0 ± 0.7) × 10⁻⁷ M restoring the enzymes' total activity (Supplementary Fig. 13). The estimated $K_D$ value indicates a high affinity for FAD, comparable to other flavoenzymes[34–36]. Flavin reactivity depends on the type of cofactor

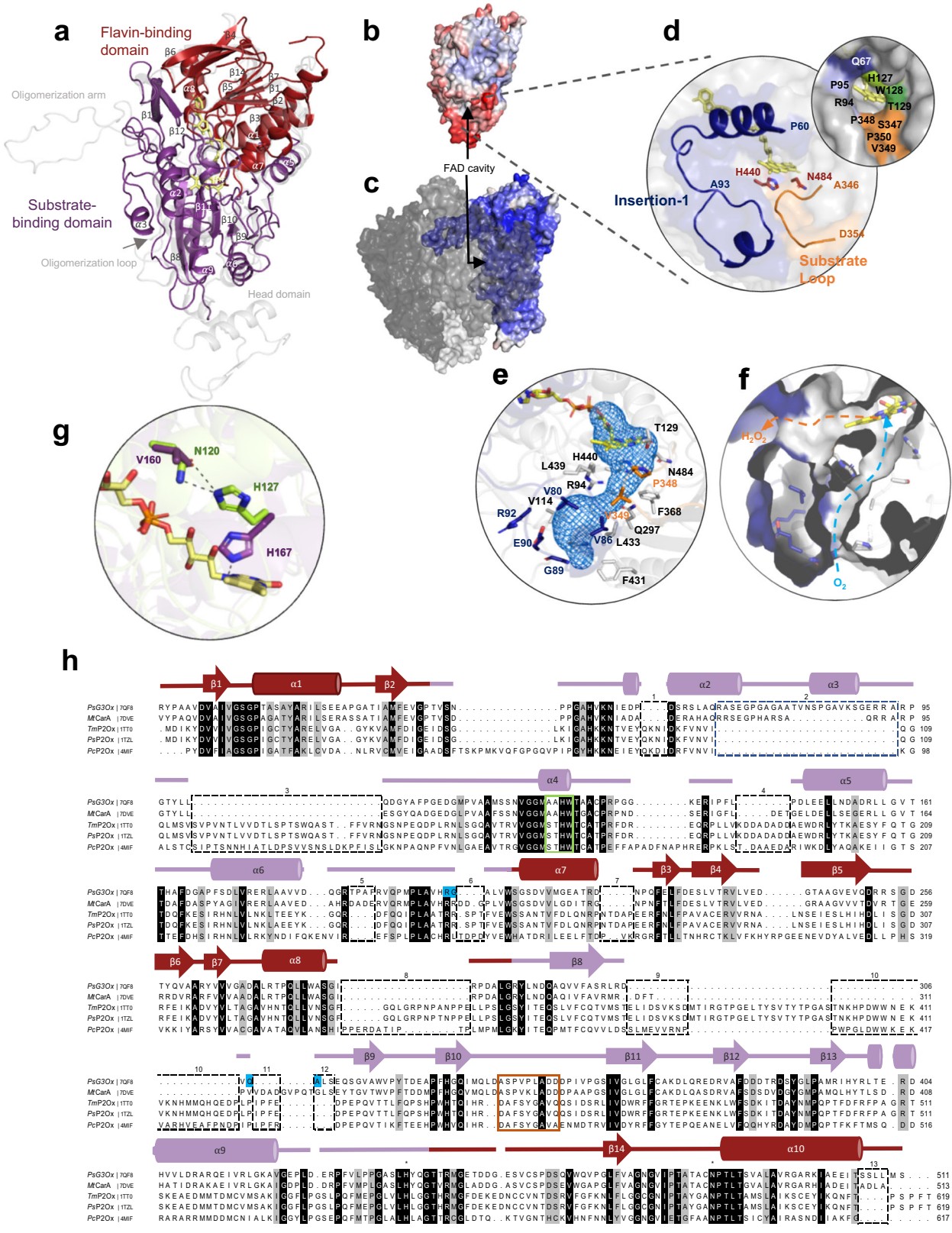

attachment[37] which is a key difference between P2Oxs and G3Oxs (Fig. 2g). To investigate whether the covalent attachment of FAD can contribute to increased reactivity of *Ps*G3Ox, (i) [125]AAHT[128] was replaced by STHW and, (ii) N120 was replaced by a Val (present in the fungal *Tm*P2Ox at that position) to remove a hydrogen bond to H127 that may preclude its covalent binding to FAD (Fig. 2g). However, variants showed non-covalently bound FAD or impaired FAD loading,

as is the case of A125S-A126T and N120V (Supplementary Fig. 14 and Supplementary Table 6).

The replacement of conserved residues H440 and N484 (Fig. 2d) by alanine resulted in inactive enzymes, confirming their key catalytic role (Supplementary Table 6); H440 is expected to act as a proton acceptor for the C2-OH group of sugar substrates, with the support of N484, that stabilizes the protonated intermediate through a hydrogen

**Fig. 2 | The structural fold of bacterial *Ps*G3Ox. a** The overall structure of *Ps*G3Ox displays the flavin- and substrate-binding domains highlighted in dark red and purple, respectively. The monomer of *Tm*P2Ox (PDB 1TT0, light gray) is super-imposed. The solvent-accessible surface of *Ps*G3Ox (**b**) and of one subunit of *Tm*P2Ox (**c**) are represented according to the *a.d.p* values, blue (6 Å$^2$) to red (107 Å$^2$). **d** Structural elements that surround the FAD cavity in *Ps*G3Ox and delimiting amino acid residues (defined using a 1.4-Å rolling probe). **e** Tunnel that connect the surface to FAD is represented in blue mesh. The delimiting residues in blue and in orange belong to the insertion-1 and substrate loop, respectively. **f** Representation of the accessible surface area that define a tunnel that can route the molecular oxygen to the FAD$^{N5}$ (blue arrow), and the active site cavity that can be the preferred pathway for the hydrogen peroxide elimination (orange arrow). **g** Comparison of

*Ps*G3Ox (residues in green) and *Tm*P2Ox (PDB 2IGK, residues in purple) flavinylation site. The H bonds are shown as black dashed lines. In all structures, the FAD is shown as sticks in yellow color. **h** Amino acid sequence alignment of bacterial and fungal POxs based on 3D superpositions of the crystal structure. The α-helices or β-chains are numbered and colored as in (**a**). Catalytic residues are highlighted with *. The flavinylation motif and the substrate loop in fungal P2Oxs are marked with light green and orange boxes, respectively. The insertions and deletions regions are highlighted with dashed boxes. The cyan-marked residues correspond to the non-visible regions. Strictly conserved amino acids are represented on black back-ground, whereas dark gray represents the most conserved residues among the selected sequences.

bond, after the hydride transfer of C2 hydrogen to the FAD$^{N5}$ atom[26,38]. *Ps*G3Ox's substrate loop, $^{346}$ASPVPLADD$^{354}$, has three polar and six hydrophobic residues and shows comparable structural flexibility to the fungal *Tm*P2Ox substrate loop ($^{452}$DAFSYGAVQ$^{460}$) (Supplementary Table 7). In the primary sequence alignment, this region is very well conserved among bacterial enzymes (Supplementary Fig. 11h), except in the case of *Ka*P2Ox, which exhibits an amino acid composition ($^{369}$DAFHYGDVP$^{377}$) comparable to the fungal enzymes. To assess the role of the loop in the catalytic properties of *Ps*G3Ox, a substrate loop truncated variant (Δloop 345–359) was constructed and characterized. This variant shows a comparable catalytic efficiency ($k_{cat}/K_m$) for D-Glc and Mang to the full-length protein, even though the productive binding is negatively affected: whereas the affinity for both substrates increases, the turnover number ($k_{cat}$) is threefold to fivefold lower (Table 1). The insertion-1 segment in *Ps*G3Ox lowers the substrate loop and active site exposure to solvent (Supplementary Tables 8 and 9). Deleting the insertion-1 loop between residues 73–93 (its most flexible region) resulted in an inactive, FAD-depleted variant. After in vitro flavinylation, a fully loaded FAD enzyme was obtained, but enzymatic activity was not recovered (Table 1). These results suggest that insertion-1 is involved in FAD incorporation and proper stabilization at the active site and plays a crucial role in catalysis.

## Conformational loop changes upon substrate binding and catalysis

The structures of complexes *Ps*G3Ox-Glc and *Ps*G3Ox-Mang are similar to the native structure with r.m.s.d. values of 0.40 Å and 0.42 Å. (Fig. 3a–c; Supplementary Fig. 15a) but display significantly (almost twofold) higher FAD cavity volumes (Supplementary Table 5). Fur-thermore, these structures show higher *a.d.p.s* values and lack visible electron density maps of residues close to the insertion-1 and substrate loop regions (Fig. 3b, c; Supplementary Figs. 15 and 16). The con-formation of these non-visible regions was modeled using Rosetta, and the loop candidates were scored based on the lowest possible Rosetta energies (Fig. 3d–f). In the *Ps*G3Ox substrate-free, similarly to *Mt*CarA, the substrate loop displays a closed conformation, and the active site has a smaller dimension and higher content of hydrophobic residues (Fig. 3d, g). In *Ps*G3Ox-Glc and *Ps*G3Ox-Mang structures, the substrate loop adopts a semi-open and open conformation, respectively. The insertion-1 loop, according to the Rosetta best models, also displays a closed and an open conformation in the substrate-free structure and substrate-complexes, respectively (Fig. 3g–i). In fungal P2Oxs, the substrate loop follows open-to-closed conformational transitions that discriminate between electron-donor and electron-acceptor sub-strates and probe the regioselective oxidation at the C2 or C3 position of monosaccharides (Supplementary Fig. 17)[39–41]. Therefore, to provide additional insights into the conformational landscape of the substrate loop and insertion-1 loop along the catalytic cycle of *Ps*G3Ox conven-tional MDs (Fig. 4k–m; Supplementary Figs. 18–21) and Gaussian accelerated MDs (GaMDs) (Fig. 4a–j; Supplementary Figs. 22–25) simulations were performed. Although the monitored variables take different values in each replicate, showing that the same initial

configuration can provide different simulation results (as expected), general trends can be observed. The A352-FAD$^{N5}$ and G84-FAD$^{N5}$ dis-tances were used to discriminate between closed (≤20 Å) and open (≥25 Å) conformations of the loops, as well as the enzyme's radius of gyration; smaller values were measured for the closed (and semi-open) states when compared to the open system. Moreover, dynamic cross-correlation analysis (DCCA) of the MD trajectories was carried out as it is a powerful tool to describe and identify functional correlated motions as well as non-active-site residues that potentially affect catalysis[42,43]. For the substrate-free Model I, the substrate loop and the insertion-1 loop remain closed during simulations (Supplementary Fig. 20), even when an enhanced sampling is pursued with the GaMD simulations (Fig. 4a, e; Supplementary Fig. 24). This observation sug-gests a clear preference for a closed active site in the absence of sub-strates, in agreement with crystallographic data. The hydrophobic composition of the substrate loop hypothetically favors this con-formation. The stability of the closed form is also apparent from the small structural fluctuations shown by the overall low root-mean-square fluctuations (r.m.s.f.) values. A colorless DCC map indicates the absence of relevant correlated motions (Supplementary Figs. 19a, 21a, 22a, e and 25a, e). On another side, when D-Glc is bound in the active site, the simulations suggest that the semi-open substrate loop con-formation (with closed insertion 1, Model II) is the only one capable of retaining D-Glc properly bound, with stable interactions with catalytic and non-catalytic residues (Fig. 4b, f, and Supplementary Fig. 26); in a semi-open substrate loop conformation there is more available space in the A346-P348 zone to accommodate D-Glc (purple triangle in Fig. 3h). In the first Model II simulation, the A352-FAD$^{N5}$ and G84-FAD$^{N5}$ distances remain ~15–20 Å (Supplementary Figs. 20 and 24). The r.m.s.f. and correlation coefficient values remain moderate (Supple-mentary Figs. 19b, 21b, 23b, f and 25b, f). However, in the second GaMD simulation, the loops adopt a more open form (distances A352-FAD$^{N5}$ > 20 Å and G84-FAD$^{N5}$ ~ 20 Å), which may relate to D-Glc move-ment inside the binding cavity, losing some of the required catalytic interactions and finally leaving the active site (Supplementary Figs. 24 and 26). Supporting the idea that the open loop conformation cannot properly retain D-Glc, in Model III simulations (open loops, with initial A352-FAD$^{N5}$ and G84-FAD$^{N5}$ distances ~30 Å), D-Glc goes away from the active site after 300 ns in cMD and before 40 ns in GaMD (Supple-mentary Fig. 26). Significantly higher r.m.s.f. values were found for insertion-1 and substrate loop that started with these open con-formations, showing the higher structural flexibility of the open state (Supplementary Figs. 19c and 23c, g). Larger correlation coefficients also appear in the DCC maps when compared to the closed systems. Remarkably, an anti-correlated motion for insertion-1 (blue long band for residues 60–90, see "Discussion" below) was observed (Supple-mentary Figs. 21c and 25c, g). Interestingly, when long GaMD simula-tions are run starting from open loops conformations in Model III*, without D-Glc in the active site, substrate loop, and insertion-1 loop follow open to closed transition (Fig. 4c, g; Supplementary Fig. 24). This agrees with the preference for closed states in substrate-free *Ps*G3Ox mentioned above. The open-to-closed transitions (Fig. 4i) are

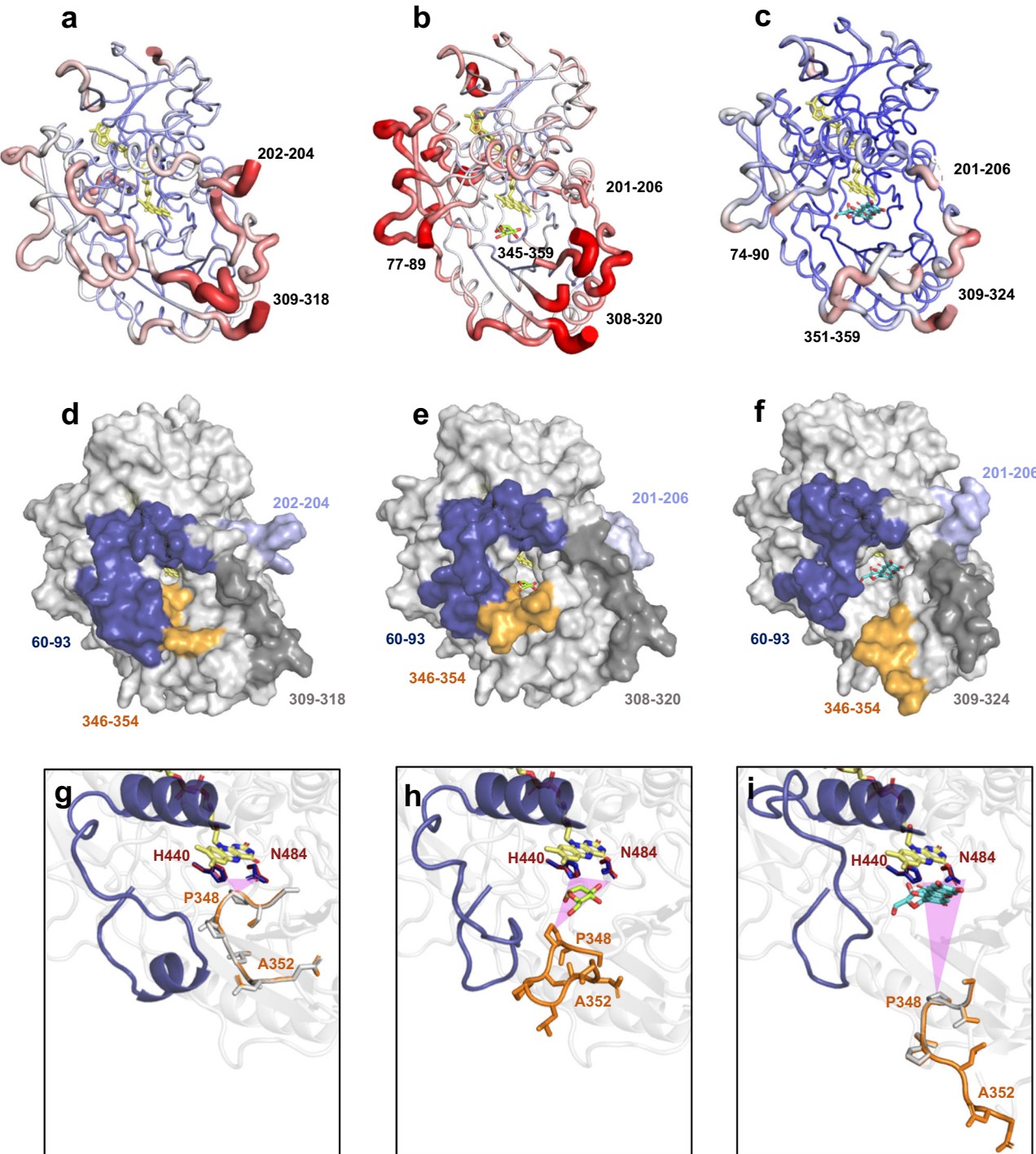

**Fig. 3 | Conformational changes of *Ps*G3Ox upon substrate binding.** X-ray structure of **a** substrate-free *Ps*G3Ox, **b** *Ps*G3Ox-Glc, and **c** *Ps*G3Ox-Mang complexes with thickness proportional to a.d.p. values, color-coded from blue (6 Å²) to red (107 Å²). Regions without electron density are highlighted near the structures. Structural models of **d** substrate-free *Ps*G3Ox, **e** *Ps*G3Ox-Glc, and **f** *Ps*G3Ox-Mang with the non-visible regions in the crystal structure modeled by Rosetta. The insertion-1 and substrate loop (including the modeled segments) are colored dark blue and orange, respectively. Cartoon representation of the active site highlighting the insertion-1 and the conformation of the substrate loop (residues 346–354) in **g** substrate-free *Ps*G3Ox, **h** *Ps*G3Ox-Glc, and **i** *Ps*G3Ox-Mang complexes. The catalytic residues are shown as sticks colored in dark red or blue for the crystal structures or models. The residues in the substrate loop are shown as sticks in gray color and orange for crystal structures and models. The purple triangles represent the interatomic distances between the catalytic pair and the residue P348 of the substrate loop. In all structures, the FAD and the substrates D-Glc and Mang are shown as sticks and colored yellow, green, and cyan, respectively.

also clear by the main movements (PC1) of the protein in this trajectory, as seen in the Principal Component Analysis (PCA) (Fig. 4d, h), and are reflected in the larger r.m.s.f. values (Fig. 4j). The conformational changes are also associated with a significant increase in (anti) correlated motions between different parts of the enzyme, which may help identify protein movements accompanying loop transitions (Supplementary Fig. 25d). Inspection of the DCC map reveals a red triangle in the bottom right, indicative of a concerted motion of all the N-terminal region (residues > 345), including most of the substrate loop. There is also a correlation (marked red band) with the movement

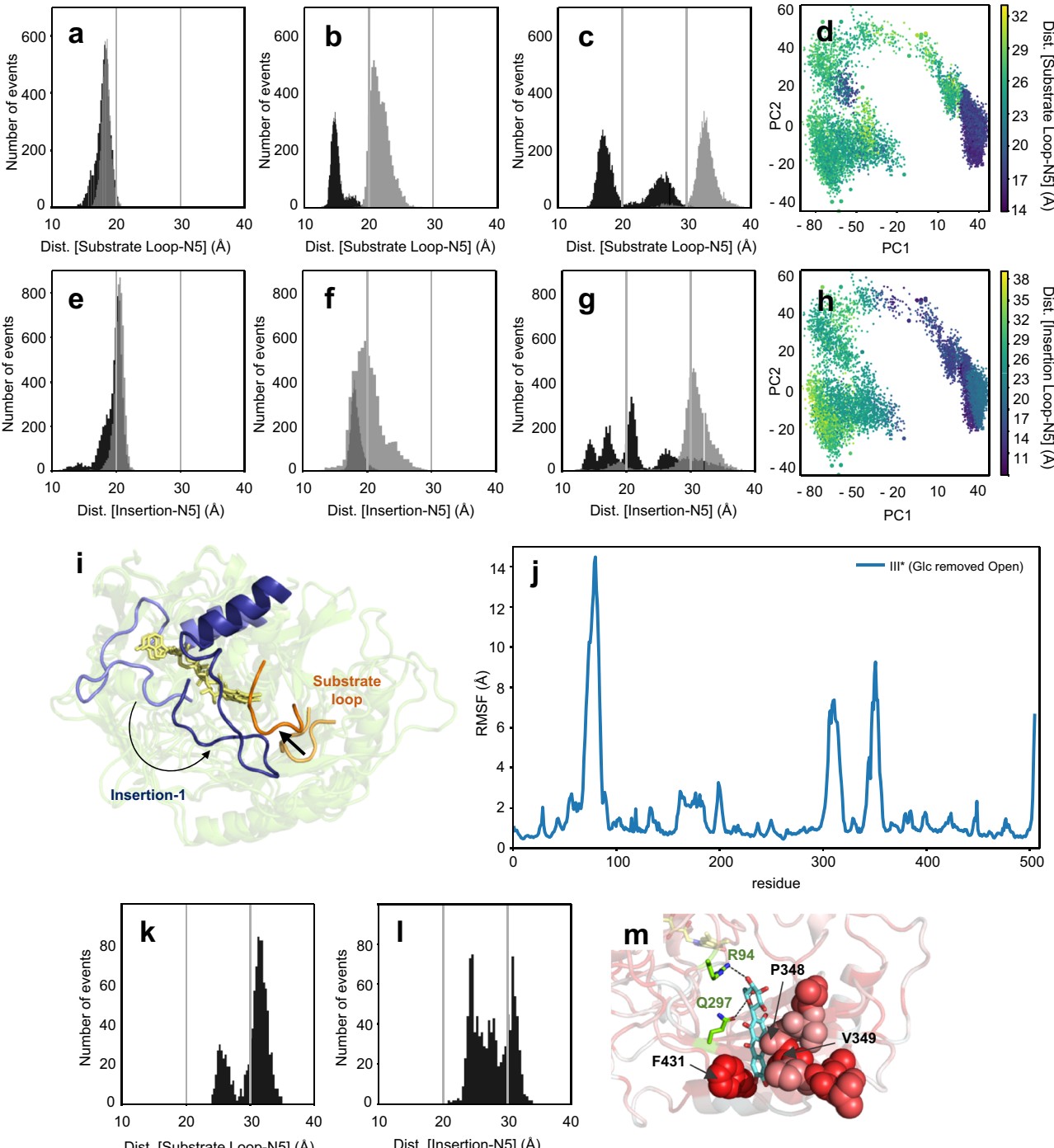

**Fig. 4 | Conformational transitions of loops close to *Ps*G3Ox active site.** Histograms of the substrate-loop distance to FAD (A352-FAD$^{N5}$) in GaMD simulations run (black and gray represent different replicates) for **a** Model I, which has no substrate and starts with a closed loops conformation, **b** Model II, which contains D-Glc and starts at semi-open conformation; and for **c** Model III*, that starts at open conformation and has no substrate. **d** Projection of the Model III* first trajectory onto the two first principal components obtained by PCA analysis, colored according to the A352-FAD$^{N5}$ distance, two main areas of structures are populated along PC1, which roughly correspond to the closed (blue) and open (green) conformations of substrate loop. Histograms of the insertion-1 distance to FAD (G84-FAD$^{N5}$) in GaMD simulations run (black and gray represent different replicates) for **e** Model I, **f** Model II, and **g** Model III*. **h** Projection of the Model III* first trajectory onto the two first principal components obtained by PCA analysis, colored accordingly to

G84-FAD$^{N5}$ distance, two main areas of structures are populated along PC1, which roughly correspond to the closed (blue) and open (green) conformations of the insertion 1. In all histograms of GaMD simulations, the black and gray colors represent two distinct simulations of 600 ns. **i** Cartoon representation of the first and last (600 ns) frames of the GaMD simulation of Model III*. The transition from open to closed state of substrate loop and insertion-1 are represented by a black arrow. FAD is colored yellow. **j** r.m.s.f. of Model III* GaMD. Histograms of the **k** substrate-loop and **l** insertion-1 distances to FAD$^{N5}$ in cMD simulation for Model IV that contains Mang and starts with open loops conformations. **m** Visual representation of the final frame (400 ns) of the corresponding simulation, where residues are colored according to hydrophobicity (white to red). Source data is provided as a Source Data file.

of residues 260−305. On the other side, the movement of all this N-terminal region is anti-correlated with that of substrate loop Pro348 (blue line) and, more importantly, with that of insertion-1 residues (*ca.* 60−93, marked blue stripe). This is in agreement with the loop movements depicted in Fig. 4i, by which the insertion-1 segment and substrate loop close towards each other and suggest that the substrate loop closure is accompanied by the displacement of the N-terminal region. Overall, the simulations performed with D-Glc suggest that the substrate loop adopts a narrow range of conformations that allow effective oxidation of this substrate, affecting D-Glc productive binding and explaining the poor catalytic parameters obtained for D-Glc. In silico molecular dynamics using Mang as substrate (Model IV) indicates that the substrate and insertion-1 loops adopt an open conformation (Fig. 4k, l) which is required to enlarge the active site pocket and allow bulkier substrates such as glycosides to bind (Fig. 3f). During the 400 ns of MD simulations of Model IV (open loops), both loops remain open (distance A352-FAD$^{N5}$ > 30 Å and G84-FAD$^{N5}$ > 20−25 Å) (Fig. 4k, l; Supplementary Fig. 20), and Mang remains correctly oriented for ~280 ns, suggesting a stable complex. However, at the end of the simulation, Mang leaves its oxidation position in the active site. Interestingly, it moves towards the substrate loop, where the mangiferin's aglycon motif is accommodated between the loop's hydrophobic residues P348 and V349 and the residue F431 (Fig. 4m). This suggests that the aglycon motif of substrates such as Mang interact with the substrate loop during substrate recruitment and/or product release. The dynamic transitions of the substrate loop and insertion-1 loop hint at a joint function of these two regions, not only in the access to the active site but also in the proper accommodation and oxidation of substrates.

### Enzyme-substrate interactions

We propose a mechanism for *Ps*G3Ox (Fig. 5a) based on the one described for sugar oxidation by P2Oxs established from density functional theory (DFT) quantum mechanical calculations and transient kinetics[26]. A hydride transfer (C−H bond breakage) from the C3 position of glycoside (C2 in the case of Glc) to the FAD$^{N5}$ occurs firstly and produces a protonated ketone intermediate (C2=O$^+$H) and the reduced flavin; this is followed by proton abstraction promoted by H440, with support of N484, that leads to the keto-sugar product. In the crystal structure of the *Ps*G3Ox-Glc complex, D-Glc$^{O2}$ is located at 4.2 and 4.7 Å of the catalytic residues H440$^{NE2}$ and N484$^{ND2}$, and at 4.2 Å from FAD$^{N5}$ (Fig. 5b). The hydrogen atom at the C2 atom of D-Glc points to the flavin N5 atom, which might facilitate a hydride transfer and support D-Glc oxidation at the C2 position[20]. In fungal P2Oxs, D-Glc establishes shorter distances (2.5−3.0 Å) to the FAD$^{N5}$ and catalytic residues[39,44]. Molecular docking of D-Glc into *Ps*G3Ox revealed a wider network of interactions than those observed in the crystal structure (Fig. 5b, c; Supplementary Figs. 27, 28 and 29). The binding of D-Glc positioned for C2, and C3 oxidation (based on the corresponding catalytic distances) was abundant (Supplementary Fig. 27). Remarkably, in the semi-open system, a higher C2/C3 ratio was obtained, suggesting a preference for oxidation of D-Glc at the C2. This observation supports that a semi-open substrate loop conformation is essential to bring and retain D-Glc close to the FAD and catalytic residues and properly orient D-Glc for C2 oxidation. Despite the significant overall sequence identity between G3Oxs and P2Oxs, the topology in the active site surrounding regions differs substantially, which point to distinct mechanisms of controlling regiospecificity. In P2Ox's (e.g., *Tm*P2Ox), C2/C3 regioselectivity is reportedly controlled by the rearrangement of the substrate loop after hydrogen bonds are established between D-Glc and residues D452 and Y456[40]. In *Ps*G3Ox, these latter residues are replaced by the hydrophobic residues A346 and P350, and no hydrogen bond interactions between D-Glc and the loop were detected in the crystallographic, MDs or docking experiments. However, and as mentioned above, docking of D-Glc to structures harboring

different loop conformations suggest that a semi-open conformation favors binding of C2 over C3 and at shorter distances to FAD$^{N5}$ and H440 (Supplementary Figs. 27 and 28), though it is not clear at the moment which type of loop-glucose specific interactions are occurring. Furthermore, the short D-Glc$^{H2}$–FAD$^{N5}$ distances sampled along the unrestricted MD simulations of the *Ps*G3Ox:FAD:Glc complex are also consistent with a preferred C2 oxidation mechanism starting with D-Glc$^{H2}$ hydride transfer (Supplementary Fig. 30a, b).

In the *Ps*G3Ox-Mang crystal complex (and dockings), more hydrogen bonds were observed, which may explain the higher affinity and catalytic efficiency of *Ps*G3Ox for this substrate compared with D-Glc (Fig. 5d, e; Supplementary Figs. 31 and 32). In agreement with the experimental trend, calculation of the differential binding free energy between Glc and Mang by MM-PBSA and MM-GBSA methods, provided an estimation of −3.6 and −4.3 kcal/mol, respectively, in favor of Mang binding (Supplementary Table 10). Notably, in both experimental and computational dockings, a clear preference for the positioning of Mang oriented towards the C3 oxidation was observed in contrast to the C2 positioning of D-Glc. The Mang$^{H3}$–FAD$^{N5}$ distances sampled along the MD simulations of the *Ps*G3Ox:FAD:Mang complex show values catalytically competent for an oxidation mechanism initiated by H3 hydride transfer, as suggested above (Supplementary Fig. 30c, d). Computational dockings were performed to explore the structural reasons behind the lack of activity toward carminic acid and rutin (Table 1; Supplementary Fig. 33a). Carminic acid binds in a non-catalytically competent manner: H$_2$/C$_2$ is relatively well oriented towards FAD$^{N5}$, but it is the OH3 that interacts with catalytic H440; no interaction with the catalytic residue N484 was observed. In the case of rutin, the dockings predict binding in an orientation compatible with catalysis at C3 (Supplementary Fig. 33b). Inhibition assays show a fivefold lower inhibition constant ($K_i$) for rutin (0.1 mM) as compared to carminic acid (Supplementary Fig. 34), suggesting that rutin can bind more strongly to the active site, supporting the dockings' data. Therefore, we hypothesize that the absence of activity with rutin might be associated with differences in its intrinsic properties, e.g., due to a hypothetic redox potential that impairs electron transfer or the lower hydrophobicity of its aglycone part preventing an optimal substrate recruitment and stabilization at the enzyme binding site. However, these possibilities were not investigated here.

Alanine mutagenesis was used to investigate the importance of residues K55, R94, T129, Q297, and Q340 in substrate binding and catalysis (Fig. 5f; Supplementary Table 6). With a few exceptions, the replacement to Ala resulted in decreased $k_{cat}$ and comparable $K_m$ values to the wild type. The absence of Q297 and, particularly, of Q340 has a significant detrimental effect in catalysis for both substrates, even if slightly higher for D-Glc. On the other hand, the oxidation of Mang is particularly affected in R94A and T129A variants, whereas the oxidation of D-Glc is primarily unaffected. These two variants, along with K55A and Q340A, display higher $K_m$ values for Mang and are thus expected to be involved in Mang's binding and productive pose in the active site. K55A shows a similar $k_{cat}$ for Mang following its furthest location from the FAD and catalytic residues in the docking simulation models (Fig. 5e), contrary to the complex crystal structure where K55 is located nearby the aglycon motif of the mangiferin (Fig. 5d). The kinetic results obtained with R94A and Q297A are in accordance with the MDs that suggest a role in anchoring bulky substrates. In *Ps*G3Ox, the residue T129 establishes hydrogen bonds with P348$^N$ and FAD$^{N5}$ (Supplementary Fig. 17a), similar to residue T169 in *Tm*P2Ox (Supplementary Fig. 17b); however, whereas T169 was suggested to trigger the transition between the different conformations of the substrate loop[39–41,44], T129 shows the same configuration independently of the substrate loop conformation (Supplementary Fig. 15e−g).

This work establishes the distinct structural, functional, and mechanistic features between two distinct phylogenetic groups with different activity profiles in the POx family of enzymes: those with a

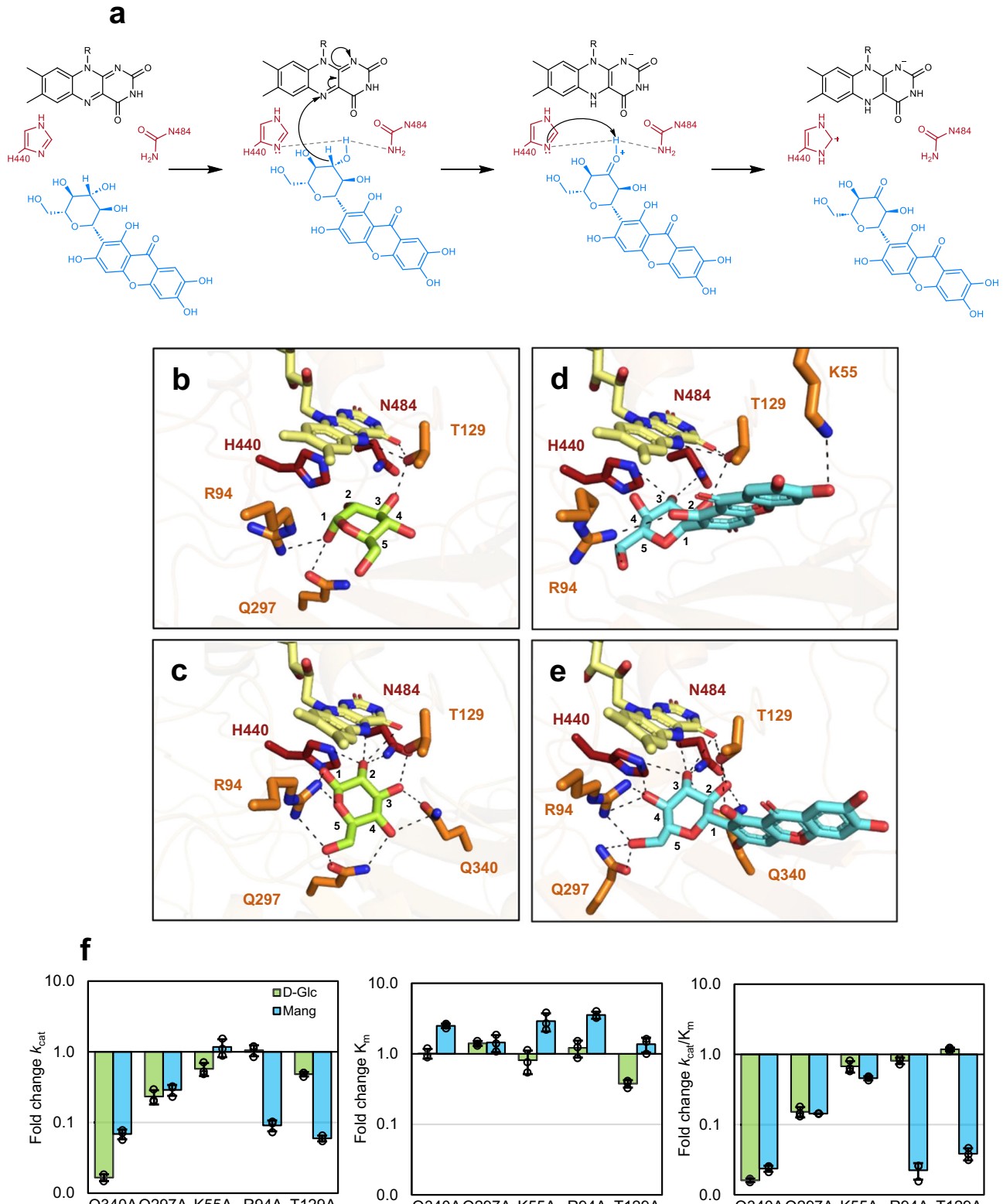

higher specificity towards the oxidation of the monosaccharide D-Glc, the P2Oxs, and those with a higher specificity towards the oxidation of the D-Glc moiety of *C*-glycosides, the G3Ox group. This work revealed that insertion-1, a striking structural segment of G3Ox's active site, plays a vital role in catalysis. Insertion-1 is located in bacterial monomeric G3Oxs at the same region where the oligomerization loop and domain is located in dimeric and tetrameric P2Ox. Therefore, we speculate that structural changes in these (non-conserved) regions

may contribute to the formation of different functional oligomeric states that might be important in modulating catalysis and substrate specificity within the POx family of enzymes. In *Ps*G3Ox (and *Mt*CarA), the substrate loop is projected into the active site cavity, sterically occluding the binding of substrates, contrary to P2Oxs fungal enzymes. This propensity to adopt a closed conformation probably relates to its hydrophobic composition and to the articulated inter-action with the insertion-1 segment promoting the access to the active

**Fig. 5 | Interactions of *Ps*G3Ox with substrates. a** Proposed reaction mechanism of the mangiferin oxidation where the proton abstraction by H440 occur after hydride transfer (adapted from Wongnate et al. [26]). Binding of D-Glc in *Ps*G3Ox-Glc crystal structure (**b**) and docking model (**c**). Binding of Mang in *Ps*G3Ox-Mang crystal structure (**d**) and in docking model (**e**). The catalytic residues (H440 and N484) are shown as sticks colored in dark-red and the non-catalytic interacting residues are colored in orange. The FAD and the substrates D-Glc and Mang are shown as sticks and colored in yellow, green, and cyan, respectively. The hydrogen bonds are shown as black dashed lines. **f** Fold-change of the catalytic parameters compared with the wild-type *Ps*G3Ox for the alanine mutants at the non-catalytic interacting residues. The catalytic parameters for D-Glc were estimated using the HRP-AAP/DCHBS coupled assay, whereas reactions with Mang were monitored by oxygen consumption in an Oxygraph. The bars and error bars represent the mean ± SD while the overlaid points represent the values for all the independent assays (*n* = 3). All reactions were performed in 100 mM sodium phosphate buffer at pH 7.5 at 37 °C. The kinetic parameters were determined by fitting the data directly on the Michaelis-Menten equation using OriginLab (see Supplementary Table 6). Triplicates were performed for all kinetic measurements. Source data is provided as a Source Data file.

site and allowing proper accommodation of the substrates. In the presence of small substrates such as D-Glc, a semi-open conformation must be adopted to create enough space for substrate binding and orientation towards C2 oxidation while simultaneously avoiding an excessive enlargement of the active-site cavity that would result in the loss of substrate, impairing the catalysis. Most likely, the difficulties in establishing this equilibrium, either by a lack of trigger(s) for the formation of the semi-open state or its maintenance, are reflected in the low specificity of G3Ox to D-Glc. In contrast, an open conformation is mandatory for the binding of Mang. In *Ps*G3Ox, substrate loop can create a "hydrophobic clamp" that interacts with the aglycone moiety of mangiferin (and presumably of other glycosides), suggesting a role of the substrate loop to assist the substrate recruitment or product release. Overall, a combined experimental and computational investigation of *Ps*G3Ox allowed mapping the relationships between the enzyme sequence-structure-function and elucidating functional transitions that accompany substrate binding and release. Such changes highlight the fine control of access to the catalytic site required by the enzyme mechanism and, in turn, the specificity offered by the enzyme towards different substrates. Work is ongoing to explore the molecular determinants for substrate specificity among bacterial POxs. This undertaking is essential for advancing fundamental biochemical insights in protein science and shedding light on the diversity of microbial catabolic pathways of natural compounds.

## Methods

### Bacterial strains, plasmids, and cultivation media

*Escherichia coli* strains, plasmids, and primers used in this work are summarized in Supplementary Table 11. *E. coli* strain DH5α (Novagen, Darmstadt, Germany) was used to propagate and amplify plasmid constructs. *E. coli* Rosetta pLysS (DE3, Novagen, Darmstadt, Germany) was used to express the wild-type *psg3ox* and its variants cloned in the pET-15b plasmid (Novagen, Darmstadt, Germany). Luria Bertani medium (LB) was used for cell cultivation, supplemented with 100 μg ml⁻¹ of ampicillin (NZYTech, Lisbon, Portugal) and, in the case of Rosetta pLysS, also with 20 μg ml⁻¹ of chloramphenicol (NZYTech, Lisbon, Portugal).

### Construction of variants by site-directed mutagenesis

The Quick-change mutagenesis protocol (Stratagene, CA, USA) was used in the construction of single variants, as well as to delete two regions of the enzyme, the insertion-1 loop between residues 73 and 93, and the substrate-binding loop between residues 345 and 359, in variant Δloop (345–359) (Supplementary Table 11). The plasmid pSM-1, carrying the wild-type *psg3ox* gene, was used as a DNA template with appropriate primers. PCRs were performed in a thermal cycler (MyCycler™ thermocycler, Biorad) in 50 μL reaction volume containing 100 ng of DNA template, 1 μM of primers (forward and reverse), and 200 μM of dNTPs (NZYTech, Lisbon, Portugal). One U of NZYProof polymerase (NZYTech, Lisbon, Portugal) was used to amplify the DNA, except for pAT27 and pAT28 where 1 U of Q5 High-Fidelity DNA polymerase (New England BioLabs, MA, USA) was used. For the single and double mutants, after an initial denaturation of 4 min at 95 °C, 25 cycles of 1 min at 95 °C, 1.5 min at 72 °C, and 10 min at 72 °C were performed, followed by a final elongation of 10 min at 72 °C. Amplification of truncated variants was performed after an initial denaturation of 30 sec at 98 °C, 35 cycles of 10 sec at 98 °C, 30 sec at 72 °C and 4 min at 72 °C, followed by a final elongation of 2 min at 72 °C. For all the PCR products, the DNA template was digested with 10 U of *Dpn*I (ThermoFisher, MA, USA) at 37 °C for 6 h, followed by purification using Illustra GFX PCR DNA kit (GE Healthcare, IL, USA). In the truncated variants, an overnight ligation with T4 ligase (ThermoFisher, MA, USA) was performed at room temperature, followed by purification with the abovementioned kit. The PCR products were transformed in *E. coli* strains using electroporation, and the presence of the desired mutation(s) or deletions was confirmed by DNA sequencing.

### Production and purification of *Ps*G3Ox

The recombinant strains *E. coli* Rosetta pLysS carrying the genes coding for wild-type *Ps*G3Ox and variants were grown in 2.5 L LB media supplemented with 100 μg ml⁻¹ of ampicillin and 20 μg ml⁻¹ of chloramphenicol in Corning® 5 L Baffled PETG Erlenmeyer flasks. The cultures were incubated at 37 °C, 100 rpm (Innova 44 incubator shaker, New Brunswick Scientific). Cultures were induced with 100 μM of isopropyl β-D-1-thiogalactopyranoside (IPTG) at an $OD_{600\,nm} = 0.8$, the temperature was lowered to 25 °C, and cells were collected by centrifugation (4420 × *g*, 10 min, 4 °C) after 16 h of cultivation. Cells were disrupted in a French press under 1000 psi, the cell debris was removed by centrifugation (18,000 × *g*, 2 h, 4 °C) and the purification of wild-type *Ps*G3Ox and variants was performed using a Histrap HP column (Cytiva, MA, USA) equilibrated with 20 mM Tris-HCl buffer, pH 7.6 with 200 mM NaCl and 10 mM imidazole. The elution was performed by applying a one-step linear gradient of 500 mM imidazole in 5 CV at a flow rate of 1 ml min⁻¹. Enzyme preparations for crystallographic trials were purified using a 1 ml-Resource-Q column (Cytiva, MA, USA) using 20 mM Tris−HCl pH 7.6 as running buffer and a gradient of 0–500 mM of NaCl for elution. Before the enzyme crystallization, the His(6x)-tag was cut from the enzyme using the thrombin cleavage kit (Abcam, Cambridge, UK) following the manufacturer protocol at 20 °C for 16 h. Afterward, the preparation was loaded on the Histrap HP column, and the flowthrough (containing the untagged enzyme) was collected. The total protein concentration was determined by Bradford assay using bovine serum albumin as standard. $Abs_{450nm}$ ($\varepsilon_{450nm}^{FAD} = 11,300\,M^{-1}\,cm^{-1}$) of purified preparations was measured to assess the functional fraction of enzyme preparations (e.g., for kinetic measurements).

### Crystallization and cryoprotection

Crystallization conditions were screened with a nanodrop crystallization robot (Cartesian, Genomic Solutions) using the sitting drop vapor diffusion method with round-bottom Greiner 96-well CrystalQuick™ plates (Greiner Bio-One, Kremsmünster, Austria). The Structure Screen I and II (Molecular Dimensions) led to the formation of *Ps*G3Ox crystals within seven days at 20 °C using 2 M ammonium sulfate and 0.1 M Tris-HCl, pH 8.5 in drops of 0.1 μl protein solution (18 mg ml⁻¹) plus 0.1 μl reservoir solution. Following this crystallization hit, microliter-scale crystals optimization proceeded using the hanging drop vapour diffusion method in XRL 24-well crystallization plates

with 500 μL of the reservoir solution (Molecular Dimensions, Newmarket, UK). Several conditions were tested: ammonium sulfate concentration ranging from 0.5 to 2 M, Tris-HCl, pH 7.0–8.5, and different ratios (1:2, 1:1, and 2:1) of protein: reservoir solution volumes. Yellow and round crystals appeared after 7–10 days, reaching dimensions of 100 μm in their three dimensions when using 2 M ammonium sulfate and 0.1 M Tris-HCl, pH 8.5, and 1 μL of protein (18 mg ml⁻¹) and 2 μL of reservoir solution at 20 °C. Crystals of the PsG3Ox-substrate complex were obtained by soaking PsG3Ox crystals in (1) the reservoir solution containing 2 M D-Glc for 30 min and (2) the reservoir solution containing 1 mM Mang for 1 min. Crystals were cryo-protected by plunging in a reservoir solution supplemented with 20% (v/v) glycerol before flash-cooling in liquid nitrogen.

### Data collection and processing

Diffraction data were measured in ID23-2 and ID30A-1 beamlines at the European Synchrotron Radiation Facility (ESRF, Grenoble, France) and in XALOC beamline at ALBA (Barcelona, Spain) for PsG3Ox, PsG3Ox-Glc and PsG3Ox-Mang complex crystals, respectively. Diffraction images of PsG3Ox were obtained with a DECTRIS PILATUS3 X 2 M detector, using 0.8731 Å radiation wavelength, crystal-to-detector distance 232 mm, and oscillations width 0.20° in a total of 360° rotation. The diffraction data of the PsG3Ox-Glc complex were obtained with a PILATUS3 2 M detector, using 0.9654 Å radiation wavelength, crystal-to-detector distance of 238 mm, and oscillations width of 0.20° in a total of 180° rotation. The diffraction data of the PsG3Ox-Mang complex were obtained with a DECTRIS PILATUS 6 M detector, radiation wavelength 0.97926 Å, crystal-to-detector distance of 553.95 mm, and oscillations width of 0.20° in a total of 180° rotation. Data were indexed and integrated with XDS[45] in a space group determined with POINTLESS[46], and the data scaled with AIMLESS[46,47]. These programs were used within the autoPROC data processing pipeline[48]. Data collection details and processing statistics are listed in Supplementary Table 4.

### Structure determination, refinement, and analysis

The three crystals belong to the same space group and show similar cell dimensions. The distribution of their Matthews coefficient[49,50] indicated a high probability of a single molecule in their asymmetric units. The phase problem of PsG3Ox was solved by molecular replacement using MORDA[51] that selected the coordinates of P. chrysosporium POx (PcP2Ox, PDB 4MIF) and Alkalihalobacillus halodurans ATP phosphoribosyltransferase regulatory subunit structure (PDB 3OD1) as search models, which led to a 99% probability solution. PHASER[52], within the PHENIX suite[53], was used to localize the two PsG3Ox-substrates structures using native PsG3Ox structure as a search model, which led to TFZ values of 35 and 42, indicating successful structures solutions[54]. Automated model rebuilding and completion were performed with PHENIX.AUTOBUILD[55] followed by manual model building performed with COOT[56], and iterative refinement cycles, using PHENIX.REFINE[53,57,58].

Structure refinement included atomic coordinates, isotropic atomic displacement parameters (a.d.p.s), and domains of translation, libration, and screw refinement of anisotropic a.d.p.s (TLS), previously defined with TLSMD server (http://skuld.bmsc.washington.edu/~tlsmd)[59]. Approximately 1.5% of reflections were randomly excluded from monitoring the refinement strategy. Solvent water molecules were automatically assigned from σA difference maps peaks neighboring hydrogen bonding acceptors/donors within 2.45–3.40 Å distances. Other solvent molecules were identified through a comparison of their shapes against electron density blobs, as well as by comparing their refined a.d.p.s with those of neighboring atoms. Some atoms were modeled with partial occupancies when hinted by the difference in Fourier maps and neighboring a.d.p.s values. As some regions of the PsG3Ox and PsG3Ox-substrate complex structures were not visible in

the electron density maps, a BUSTER protocol was applied to search for missing atoms[60,61]. The stereochemistry of the refined structures was analyzed with MOLPROBITY[62]. Three-dimensional superposition of polypeptide chains was performed with MODELLER[63]. The tunnel and cavity were determined using the MOLE[64] and DoGSiteScorer[65], respectively. Figures of structural models were prepared with PyMOL[66,67]. Refinement statistics are presented in Supplementary Table 4. Structure factors and associated structure coordinates of PsG3Ox, PsG3Ox-Glc, and PsG3Ox-Mang complex were deposited in the Protein Data Bank (www.rcsb.org) with PDB codes 7QF8, 7QFD, and 7QVA, respectively.

### High-pressure krypton gas

PsG3Ox crystals high pressurization with krypton (Kr) was performed at ESRF (Grenoble, France) in order to localize hydrophobic tunnels[68,69]. Crystals were harvested with a specific pluggable sample support and transferred into a high-pressure cooling system. After 3 min of pressurization under krypton gas at 220 bar, the crystals were dropped into the bottom of the tube and directly flash-cooled. Finally, the system was depressurized and the crystals were handled in liquid nitrogen for data collection at beamline ID30B[70]. A dataset from the PsG3Ox crystal was collected to a resolution of 2.35 Å using a wavelength of 0.8610 Å, in order to use the Kr anomalous signal to validate its location in the crystal structure. Data processing was performed with autoPROC pipeline[48] similar to the non-pressurized PsG3Ox crystals. The crystal structure of PsG3Ox pressurized with krypton was solved by molecular replacement with Phaser[52] using the PsG3Ox structure as a search model. The crystal structure was validated using an anomalous difference Fourier map calculated from a dataset collected above the K absorption edge of Kr (14.2 keV). Refinement cycles were tested by reducing successively the Kr occupancy to 0.68, where its a.d.p. reached values similar to the neighboring atoms.

### Loop modeling

Rosetta loop modeling was used to build the non-visible loops in the PsG3Ox (202–204 and 309–318), PsG3Ox-Glc (77–89, 201–206, 308–320, and 345–359), and PsG3Ox-Mang (74–90, 201–206, 309–324 and 351–359) complex structures. Loop regions were generated with the cyclic coordinate descent (CCD) loop closure algorithm[71] using fragments from proteins with known structure[72] a full-atom refinement step was performed with the next-generation kinematic (NGK) closure robotics-inspired conformational sampling protocol[73]. The crystal structures of PsG3Ox, PsG3Ox-Glc, and PsG3Ox-Mang were kept intact except for the loop regions that were created, with repacking of the side chains within 10 Å of the remodeled region. A total of 500 loops were built, and to find the best loop candidate each model was scored by its Rosetta energy score and contacts with the substrate molecules.

### Prediction of structural models of bacterial POxs by AlphaFold

To investigate the structural differences between the bacterial POxs, models of the bacterial enzymes AgCarA, CarA, ScP2Ox and KaP2Ox were predicted using AlphaFold that was accessed on 22 November 2022 through the Google Colab platform and AlphaFold2_advanced option https://colab.research.google.com/github/sokrypton/ColabFold/blob/main/beta/AlphaFold2_advanced.ipynb#scrollTo=Riekgf0KQv_3[74]. The refinement was performed using the Amber-relax option to enhance the accuracy of the side chains geometry. The predicted local-distance difference test (pLDDT) confidence values (higher = better) are indicated in the B-factor column. Only the best model among the five best given by default was examined.

### In vitro deflavinylation and reconstitution of PsG3Ox

Apo-PsG3Ox enzyme preparation was obtained following an immobilized unfolding/refolding procedure. A FAD-loaded preparation

*Ps*G3Ox was immobilized onto a 5 ml-HisTrap HP column (Cytiva, MA, USA) pre-equilibrated with 20 mM Tris-HCl, pH 7.6 (supplemented with 0.2 M of NaCl), and a urea gradient (0–1.3 M in 8 column volumes (CV)) was applied using 0.5 ml min⁻¹ as flow rate. After the cofactor release, as monitored by absorbance at 450 nm in the AKTA system, a refolding gradient was applied to decrease the urea concentration from 1.3 M to 0 M in 18 CV, followed by 5 CV of urea-free buffer to remove all denaturants from the column. The apo-enzyme was eluted using a gradient of 0–1 M of imidazole in 5 CV, and the imidazole was removed with a PD-10 desalting column (GE Healthcare, IL, USA). To assess the secondary structure, far-UV circular dichroism spectra were performed in Jasco J-815 using a protein concentration of 0.15 mg ml⁻¹ with a bandwidth of 2 nm, DIT: 8 s, Scan speed: 50 nm min⁻¹. The tertiary structure was also checked by following the intrinsic fluorescence emission of *Ps*G3Ox tryptophan when excited at 296 nm in a Cary Eclipse fluorimeter. To reconstitute the holo-*Ps*G3Ox and determine the affinity constant to the FAD cofactor, the apo-*Ps*G3Ox was incubated for 1 h with several ratios of flavin adenine nucleotide (FAD, Sigma Aldrich, MO, USA) at room temperature. The quenching of intrinsic fluorescence of the tryptophan's that are localized close to the cofactor was used to follow the binding of the FAD to the protein. An emission spectrum (310–450 nm) was recorded when the sample was excited at 296 nm in all ratios apo-*Ps*G3Ox: FAD. The inner filter effect from FAD absorbance was corrected using $F_{corr} = F_{obs}\,\text{antilog}(\frac{Abs_{ex}+Abs_{em}}{2})$ ($F_{corr}$ is the fluorescence corrected to the inner filter effect, $F_{obs}$ is the measured fluorescence, $Abs_{ex}$ is the absorbance of the sample at the excitation wavelength: 296 nm, and $Abs_{em}$ is the absorbance of the sample at each emission wavelength. The absorbances were measured in the FAD solution without enzyme). The total fluorescence was determined by integrating the corrected emission spectra. The experimental protein–ligand bond fraction given by the normalized fluorescence as a function of ligand/protein ratio was fitted to the one-binding site equation: $Y = \frac{(1+\frac{L_T}{P_T}+\frac{K_D}{P_T})-\sqrt{(1+\frac{L_T}{P_T}+\frac{K_D}{P_T})^2-4\frac{L_T}{P_T}}}{2}$[75] ($Y$ is the protein-ligand bound fraction, $L_T$ is the total concentration of the ligand, $P_T$ is the total protein concentration and $K_D$ is the dissociation constant). $K_D$ was determined using the Solver ad tool of Excel to minimize the differences between the experimental points and the fit equation. To test the activity of the reconstituted *Ps*G3Ox, the incubated enzyme preparations with several ratios of Enzyme:FAD were tested towards ᴅ-Glc oxidation in a reaction mixture containing 0.5 M of ᴅ-Glc, 0.1 mM 4-Aminoantipyrine (AAP, Acros organics, Geel, Belgium), 1 mM 3,5-dichloro-2-hydroxybenzenesulfonic acid sodium salt (DCHBS, Alfa Aesar, MA, USA) and 8 U ml⁻¹ Horseradish peroxidase (HRP, PanReac Applichem, Darmstadt, Germany). The absorbance at 515 nm was monitored in a Synergy2 microplate reader (BioTek, VT, USA).

### Optimization of activity assay for *Ps*G3Ox

To optimize the oxidase assay to measure the activity of *Ps*G3Ox for sugars, the chromogens ABTS⁺⁺ and N-(4-antipyryl)-3-chloro-5-sulfonate-p-benzoquinone-monoimine which are formed by the oxidation of 2,2′-azino-bis(3-ethylbenzothiazoline-6-sulfonic acid) (ABTS, ApplieChem, Darmstadt, Germany) or AAP + DCHBS in the coupled assay upon HRP activity were tested. ABTS⁺⁺ was generated by oxidation of 10 mM ABTS using 1 U ml⁻¹ of CotA laccase[76]. CotA laccase was removed from the reaction mixture by ultrafiltration using an ultra-centrifugal filter vivaspin20 of 30 kDa cutoff (Cytiva, MA, USA) at 4000 rpm for 30 min (Eppendorf 5810R centrifuge). The compound N-(4-antipyryl)-3-chloro-5-sulfonate-p-benzoquinone-monoimine was generated by mixing 0.1 mM AAP, 1 mM DCHBS, 8 U ml⁻¹ HRP and 0.1 mM H₂O₂. HRP was removed by ultrafiltration using an ultra-centrifugal filter vivaspin20 of 30 kDa cutoff (Cytiva, MA, USA) at 4000 rpm for 30 min (Eppendorf 5810R centrifuge). The capability of

*Ps*G3Ox to use these compounds as final electron acceptors was monitored spectrophotometrically in a reaction mixture containing 1 M of ᴅ-Glc, 35 µM of chromogen, and 10 µg of wild-type *Ps*G3Ox. The reduction of the chromogens was tested in aerobiose or anaerobiose by bubbling for 15 min the reaction mixture with oxygen or nitrogen, respectively. The absorbance at 420 nm for ABTS⁺⁺ or 515 nm for N-(4-antipyryl)-3-chloro-5-sulfonate-p-benzoquinone-monoimine compound was measured over time in a Nicolet Evolution 300 spectrophotometer (Thermo Industries, MA, USA).

### Apparent steady-state kinetics

Apparent steady-state kinetics measurements were performed at 37 °C in 100 mM sodium phosphate buffer, pH 7.5, and reactions were started with the addition of enzyme. The kinetic parameters for ᴅ-glucose (ᴅ-Glc, PanReac Applichem, Darmstadt, Germany), ᴅ-galactose (PanReac Applichem, Darmstadt, Germany), ᴅ-ribose (VWR, PA, USA), ᴅ-xylose (Sigma Aldrich, MO, USA), and ʟ-arabinose (Sigma Aldrich, MO, USA) were measured using coupling assay containing 0.1 mM AAP, 1 mM DCHBS, 8 U ml⁻¹ HRP and different concentrations of substrate. Enzymatic activity was monitored using a Synergy2 microplate reader (BioTek, VT, USA) following the formation of N-(4-antipyryl)-3-chloro-5-sulfonate-p-benzoquinone-monoimine (a pink chromogen) at 515 nm ($\varepsilon_{515} = 26{,}000\,M^{-1}\,cm^{-1}$). The kinetic parameters for molecular oxygen were measured in an Oxygraph system (Hansatech instruments, Pentney, UK) to follow oxygen consumption in reactions containing 1 M ᴅ-Glc as an electron donor and different oxygen concentrations pre-set by bubbling O₂ or N₂ gas. The oxidation of the glycosides mangiferin (Sigma Aldrich, MO, USA), rutin (Acros Organics, Geel, Belgium), and carminic acid (Sigma Aldrich, MO, USA) were followed by oxygen consumption in the Oxygraph apparatus in reactions containing 0 - 2 mM of Mang, 0–0.5 mM of rutin or 0 – 1 mM of carminic acid. Specific activity was calculated considering the preparation's functional (FAD-loaded) enzyme ratio. Apparent steady-state kinetic parameters ($k_{cat}$ and $K_m$) were determined by fitting data directly into the Michaelis-Menten equation using Origin-Lab software. For inhibition assays, the steady-state kinetics for ᴅ-Glc were performed as described below in the presence of 0–0.5 mM rutin or 0–0.2 mM carminic acid. The data was represented using a Lineweaver Burk plot and the inhibition constants were estimated based on a secondary plot of the slopes against inhibitor concentration.

### Transient-state kinetics

Transient-state kinetics were conducted inside an anaerobic glove box to ensure an oxygen-free atmosphere at 25 °C in 100 mM sodium phosphate buffer, pH 7.5 supplemented with 0.2 M of NaCl. For the reductive half-reaction, the deoxygenized *Ps*G3Ox enzyme preparation was mixed with ᴅ-Glc or Mang solutions (prepared anaerobically in the same buffer) in a Hi-Tech SF-61DX2 stopped-flow apparatus (TgK Scientific, Bradford-on-Avon, UK) coupled to a diode array. The reduction of enzyme cofactor was followed by the decrease of absorbance at 460 nm during time. For the oxidative half-reaction, the *Ps*G3Ox enzyme (previously reduced by 50 mM of ᴅ-Glc) was mixed with buffer containing different O₂ concentrations (previously prepared by mixing in different ratios the oxygenized and deoxygenized buffer). The reoxidation of *Ps*G3Ox by dioxygen was followed in a stopped-flow apparatus coupled to a diode array following the increase of absorbance at 460 nm over time. To determine the real oxygen concentration used to re-oxidize the enzyme, the oxygen containing buffers were mixed with 300 µM sodium dithionite (Sigma Aldrich, MO, USA) and its oxidation through the following reaction: Na₂S₂O₄ + O₂ + H₂O → NaHSO₄ + NaHSO₃ was monitored by the decrease of the absorbance maximum of sodium dithionite ($\varepsilon_{315} = 8043\,M^{-1}\,cm^{-1}$) in the stopped-flow apparatus under a photomultiplier detector (TgK Scientific, Bradford-on-Avon, UK). All oxidative and reductive half-reactions traces were analyzed using Kinetic

studio software (TgK Scientific, Bradford-on-Avon, UK), and the data were adjusted to a single exponential $[P] = -A \times \exp^{-k_{obs} \times t} + C$ (where $[P]$ is the cofactor concentration in the enzyme preparation, $A$ is the amplitude of the exponential extrapolated to infinite time, $k_{obs}$ is the pseudo-first-order constant, t is the time of reaction and $C$ is a constant equal to the absorbance at 460 nm at an infinite time) using the Solver Ad tool of Microsoft excel.

## Identification of mangiferin oxidation product

Oxidation of Mang was performed under aerobic conditions at 25 °C, pH 7.5 in 30 mL of Milli-Q water containing 20 mg of Mang, 1 U ml$^{-1}$ of Catalase (Sigma Aldrich, MO, USA), and 1 U ml$^{-1}$ of PsG3Ox. To estimate the time needed to have a high yield of oxidized Mang, a time-course of the reaction was performed in a thin layer chromatography (TLC) on silica gel 60 F254 sheet (Merck, Darmstadt, Germany) using a mixture of butanol, acetic acid, and water in the proportions 4:1:2.2 (v/v) as mobile phase. The TLC was revealed using the diphenylamine-aniline-phosphoric acid reagent[77], a system used to distinguish sugars. For the NMR characterization, the reaction occurred for 30 min, and then the enzymes were removed by ultrafiltration using a vivaspin20 of 30 kDa cutoff (Cytiva, MA, USA). The water in the mixture was evaporated under low pressure on a rotary vacuum evaporator, and the resulting sediment was resuspended in ~ 600 μl of dimethyl sulfoxide-d6 (Merck, Darmstadt, Germany). The Mang and the reaction product, both in DMSO-d6, were analyzed through $^1$H, $^{13}$C APT, COSY, and HMQC NMR in a Bruker Avance II + 400. $^1$H NMR spectra were obtained at 400 MHz and $^{13}$C at 100.61 MHz. The spectra were analyzed using Bruker topspin 3.2 software.

## Molecular dynamics simulations

Molecular dynamics simulations (MD) of PsG3Ox were carried out to further explore the conformational preferences of the enzyme at different stages of the catalytic cycle. First, MD simulations of 400 ns production runs were performed of four systems built as follows: (i) Model I: taken from the PsG3Ox crystal structure (without substrates), which shows closed substrate and insertion-1 loops; (ii) Model II: obtained from the same crystal structure but with the substrate loop sampled with the Yasara Sample Loop function and D-Glc docked (see below) in a binding mode compatible with C2 oxidation; this model presents a semi-open substrate loop and a closed insertion-1 loop; (iii) Model III: prepared from the PsG3Ox-Mang crystal structure, with Mang removed, the missing parts of the loops were built with the Yasara Build Loop function, and D-Glc docked in a C2 binding mode; this model shows open substrate and insertion-1 loops; (iv) Model IV: derived from the PsG3Ox-Mang crystal structure (in a binding mode compatible with C3 oxidation) by building the missing parts of the open substrate and insertion-1 loops. Missing hydrogen atoms were added, and the protonation state of the titratable residues was assigned with the Yasara hydrogen bond networks optimization and pKa prediction tools at pH 7[78]. The systems were solvated with a solvation box and neutralized with NaCl. The conventional MDs, cMDs, were set up and run using Yasara[79]; the AMBER14 force field[80] and TIP3P water model[81] were used. A two-step equilibration was carried out for 400 ps: first, the system's temperature was increased in 10 steps from 30 K up to 300 K, followed by a second step with constant temperature and box dimensions. The bonds and angles involving a hydrogen atom were fixed. A restrain on all non-hydrogen atoms of the complexed protein-ligand was applied so that the equilibration would mainly affect the system's solvent. Bonded and non-bonded forces were updated every 2 and 5 fs, respectively. The protein-ligand complex was then released, and a production step of 400 ns was carried out. Gaussian accelerated MDs (GaMDs)[82] were then run for Model I–III to access effective longer simulation times and wider conformational explorations, especially of the substrate loop. The starting structures for these GaMD simulations were the 100 ns structure from

the corresponding classical MDs. An extra model (Model III*) was built from Model III by removing the bound D-Glc substrate to see if loop transitions could be observed. The GaMD simulations were run with the AMBER 20 program[83]. The GaMD protocol consisted of an initial equilibration stage where the potential boost was applied, boost parameters were updated, and production runs were updated with fixed boost parameters. A dual boost on dihedral and total potential energy was applied (igamd = 3). Two simulations of 600 ns were run for each Model I, II, and III*, whereas the simulation for Model III was stopped after 200 ns as D-Glc was observed to leave the active-site. GaMD inputs were generated following the recommendations from the developers[84]. The convergence of all simulations was assessed by calculating the RMSD values of the protein C$_{alpha}$ atoms. The distances between FAD$^{N5}$ and A352, as well as G84, and the protein's radius of gyration were used as an indicator of the conformational state of the substrate and insertion-1 loops, respectively. Root-mean-square fluctuations (r.m.s.f.), Principal Component (PCA) and dynamic cross-correlation (DCCA) analyses were used to further describe the loop's conformational landscape and dynamics. More details about the systems' setup used for each of the models built to run MD can be found in Supplementary Table 12.

## Protein-ligand docking

Protein-ligand docking calculations were carried out on different protein structures from the cMD trajectories using AutoDock VINA as implemented in YASARA.[71,85,86] D-Glc was docked in PsG3Ox's active site among 100 frames from semi-open (Model II) and open (Model III) substrate's loop MDs, respectively. On the other hand, Mang was docked among 100 frames from the open substrate loop MD (Model IV). The rest of the substrate's loop conformations, particularly those with the closed loop, did not allow proper positioning of the substrate in the active site due to steric clashes, and they were discarded. D-Glc was docked using the YAMBER force field[87] on a 20 × 20 × 20 Å cuboid docking cell centered on FAD$^{N5}$, while Mang was docked on a 34 × 34 × 34 Å cuboid docking cell centered on the same atom. A total of 16 ligand conformations were generated per frame, potentially yielding 1600 ligand-enzyme combinations per ligand. Distances between the glycoside's D-Glc group and FAD$^{N5}$ (N5-HC2; N5-HC3), as well as between H440 (NE2-HO2; NE2-HO3), were measured to filter structures in potential catalytically relevant C2 (NE2-HO2 and N5-HC2 < 4 Å) and C3 (NE2-HO2 and N5-HC2 < 4 Å) binding modes that could go into an optimization protocol, with a minimization step of the whole complex. The initial (pre-optimization) and final distances between FAD$^{N5}$/H440 and the D-Glc group of each glycoside, as well as the total energies of the entire system and the ligand binding energies, were obtained for further analysis. The minimum distances between the ligand and residues Q297, Q340, R94, T129, K55, were measured. The number of events (frequency) for each measured distance and energy, depending on the ligand-binding mode and substrate loop conformation, were plotted in different histograms. Rutin and carminic acid were docked following the same protocol as for Mang.

For D-Glc and Mang, docking solutions compatible with catalysis were selected and submitted to unrestricted cMDs attaining 100ns-long production runs (following previous 1.1 ns of equilibration). Ten replicates were performed for each substrate in order to sample more enzyme:substrate complexes. The analysis of the trajectories with the MMPBSA.py[88] module allowed the estimation of the binding free energy for each substrate, as well as their relative value, by the Molecular Mechanics/Poisson–Boltzmann Surface Area (MM-PBSA) and the Molecular Mechanics/Generalized Born Surface Area (MM-GBSA) methods.

## Reporting summary

Further information on research design is available in the Nature Portfolio Reporting Summary linked to this article.

## Data availability

The structural coordinates generated in this study have been deposited in the Protein Data Bank under accession codes: 7QF8, 7QFD, and 7QVA. The biochemical data generated in this study are provided in the Supplementary Information/Source Data file. The structural data used in this study are available in the Protein Data Bank database under accession code 7DVE, 1TT0, 4MIF, 2IGK, 3OD1 and 1TZL. The molecular dynamics input files, as well as the initial and final coordinates of each simulations run are provided in a Supplementary Data 1. Source data are provided with this paper.

## Code availability

All computer analysis scripts used for molecular dynamics and molecular docking are available on GitHub (https://github.com/insilichem/utils_PsG3Ox/)[89].

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

## Acknowledgements

We thank Diana Santos for preliminary data, Teresa Catarino with stopped-flow analysis, Tiago N. Cordeiro for help with Rosetta, Philippe Carpentier for support with krypton high-pressure experiments, Pedro Matias and Maximino Manzanera for valuable discussions. We thank the beamline staff at ESRF (Grenoble, France) and ALBA (Barcelona, Spain) for their support during the synchrotron data collection and Teresa Silva and Cristina Timóteo (Research Facilities, ITQB-NOVA) for technical assistance. The NMR data were acquired at CERMAX, ITQB-NOVA, Oeiras, Portugal, with equipment funded by FCT, project AAC 01/SAICT/ 2016. This work was supported by the Fundação para a Ciência e Tecnologia, Portugal, grants, 2022.02027.PTDC (L.O.M.), MOSTMICRO-ITQB (UIDB/04612/2020 and UIDP/04612/2020) (L.O.M. and M.R.V.), LS4FUTURE Associated Laboratory (LA/P/0087/2020) (L.O.M. and M.R.V.), PTDC/BII-BBF/29564/2017 (L.O.M.), UIDB/04326/2020, UIDP/ 043226/2020 and LA/P/0101/2020 (EPM) and FCT PhD fellowships 2020.07928 (A.T.), 2022.13872 (T.F.), and 2022.09426 (M.V.R.). B-Ligzymes (GA 824017) from the European Union's Horizon 2020 Research and Innovation Program is also acknowledged for funding T.F. secondment at Zymvol and F.S. secondment at ITQB NOVA. L.M. and X.F.L. acknowledge PID2021-126897NB-I00 project and PRE2019-088412 fellowship, funded by MCIN/AEI/10.13039/5011000 11033/ FEDER, EU.

## Author contributions

L.O.M. conceived this study; A.T. performed the mutagenesis, kinetic, and biochemical characterization of enzymes; E.P.M. helped with deflavinylation and flavinylation experiments; T.F., P.B., and C.F. solved the crystal structures; M.V.R and M.R.V. characterize reaction products; T.F, X.F.L., M.F.L, F.S, and L.M. performed MD simulations, docking, and computational analysis. All authors contributed to the experimental design, interpretation of results, and writing the paper.

## Competing interests

The authors declare no competing interests.
