## [Peer Review File · Nature Communications]

Reviewers' Comments:

Reviewer #1:

Remarks to the Author:

The manuscript by Taborda et al. reports identification of substrate specificity, regio-specific oxidation and crystal structure of a new enzyme PsGO3x, a member of the GMC flavoenzyme oxidases. Authors carried experiments with various sugars and glycosides to measure kinetics parameters of the enzymes with these compounds. The enzyme sequence was analyzed its evolutionary relationship with other members of the GMC family. The enzyme crystal structures were solved and MD simulations were carried out to obtain mechanistic insights into the control of substrate recognition.

Overall, this work contributes to enhancing the understanding of how PsGO3x controls the selection of -C-H bond oxidation and also provides useful structural and functional data for the enzyme of this class. However, I feel that the data reported are not very surprising and not sure if this work would appeal to broad readership. The work would fit better with a specialized journal.

The authors may consider these issues to improve the manuscript.

Page 3, transient kinetics is important information to understand mechanisms of this type of enzymes. The authors should present these data in the main manuscript.

Sequence identity of around 60% to other well-studied, known enzymes is quite significant. The authors should discuss and compare the similarity and difference for controlling regio-specificity of sugar oxidation with other enzymes such as pyranose 2-oxidase.

Reviewer #2:

Remarks to the Author:

The work elucidates the crystal structure of the novel bacterial Glycoside 3-Oxidases and provides experimental insight into substrate binding with the help of mutagenesis and some MD studies. The study is novel and could be published in Nature Commun. however some points of revision should be addressed:

- i) Regarding the MD simulations - I can not see the all relevant RMSD plots, RMSF plots and Radius of Gyration plots. The current plan MD trajectory is short ideally should be 1 microsecond, nonetheless that accelerated MD is later performed.
- ii) the binding free energies should be calculated at least with MMPBSA/GBSA and correlated to the experiment
- iii) the MD and Gaussian-accelerated MD analysis should have been more comprehensive. The authors should explore and analyze in a context the correlated motions of the different system. Some current references can also be used as a guidance and cited on such analysis e.g.
<https://chemistry-europe.onlinelibrary.wiley.com/doi/full/10.1002/cphc.202200649>
<https://pubs.acs.org/doi/10.1021/jacsau.2c00345>
- iv) also ideally the reaction mechanism should have been explored using QM/MM based on the MD snapshots. For a publication in Nature Commun. this should be expected as well. At least authors should comment on this and why it is not done.

Reviewer #3:

Remarks to the Author:

Taborda et al have identified a new glycoside 3-oxidase involved in C-glycoside metabolism and conducted biochemical and structural biology studies on it. The biochemical characterization results provide scientifically robust insights into the function and properties of PsG3Ox. However, additional data is required to support the crystallography experiment results, particularly with regard to substrate binding. Overall, the study provides substantial information regarding the function and mechanism of action of glycoside 3-oxidase. However, additional data is necessary to

support the experimental results.

1. Validation report suggests that the occupancy of the substrates in the complex structure is not high.

1) For instance, the electron density for D-glucose could not be confirmed in the PDB validation report, and relevant electron density maps should be added to the supporting data.

2) Additionally, there was no negative mFo-Fc observed for mangiferin in the PDB validation report, and a discussion on this is needed. An electron density map for mangiferin should also be added to the supporting data.

2. Electron density maps for amino acid regions 60-93 and 346-354, which show large conformation changes depending on the substrate, should be included in the supporting data.

3. The quality of the sequence alignment figure needs to be improved as well.

4. The authors also need to describe the oligomeric state of PsGO3x obtained experimentally and its crystal structure, as they emphasized its importance.

5. Finally, the methods section is missing references to crystallography programs such as MolProbity and Modeller, and the reference section lacks page information for several papers.

Reviewer #4:

Remarks to the Author:

In this manuscript, Taborda et al present the kinetic and structural characterization of a flavin-dependent oxidoreductase that performs an alcohol to ketone oxidation reactions for sugars appended to flavonoids, among other molecules. The reviewer has the following comments for the authors to consider:

1. Brevity and clarity - considering the structures for founding members of this enzymes class are already available, the structure description can be significantly shortened and streamlined. Same for the kinetic description. There is no question that the activity of the enzyme is lower for monosaccharides alone, this can be brought out in a much more simply.

2. Is there a rationale for attempting to "trigger a covalent attachment of FAD" to the enzyme? The motivation for performing this experiment is not clear, and as above, the description of FAD binding can be shortened.

3. If molecular oxygen is the terminal electron acceptor, does the structure inform the mode/site of oxygen binding? How does oxygen access the reduced flavin?

4. Together with Figure 5 where the binding of the substrates relative to the flavin cofactor is illustrated, it would be helpful to see a rendering of the proposed reaction mechanism with the hydride transfer step formally illustrated. It would then be useful to discuss the substrate binding modes relative to the flavin-N5.

5. Minor comment - some of the axis labels in Figure 4 are too small to be legible.

ANSWERS TO REVIEWERS

Reviewer #1

The manuscript by Taborda et al. reports identification of substrate specificity, regio-specific oxidation and crystal structure of a new enzyme PsGO3x, a member of the GMC flavoenzyme oxidases. Authors carried experiments with various sugars and glycosides to measure kinetics parameters of the enzymes with these compounds. The enzyme sequence was analyzed its evolutionary relationship with other members of the GMC family. The enzyme crystal structures were solved and MD simulations were carried out to obtain mechanistic insights into the control of substrate recognition.

Overall, this work contributes to enhancing the understanding of how PsGO3x controls the selection of -C-H bond oxidation and also provides valid structural and functional data for the enzyme of this class. However, I feel that the data reported are not very surprising and not sure if this work would appeal to broad readership. The work would fit better with a specialized journal.

The authors may consider these issues to improve the manuscript.

Page 3, transient kinetics is important information to understand mechanisms of this type of enzymes. The authors should present these data in the main manuscript.

ANSWER:

We have considered the reviewers' suggestions, and to bring the transient-state kinetics data to the main manuscript, we have included a new panel (b) in Fig 1. Furthermore, transient kinetic experiments using Mang as the electron donor were also included as follows (p. 4): *"Additionally, transient state kinetics were also performed to estimate the pseudo-second-order constants for the reductive-half reaction using Mang as substrate, $(0.27 \pm 0.02) \times 10^5 M^{-1}s^{-1}$, further supporting that Mang is a preferred substrate than D-Glc (Fig. 1b, Supplementary Figure 1)."*

Sequence identity of around 60% to other well-studied, known enzymes is significant. The authors should discuss and compare the similarity and differences in controlling the regio-specificity of sugar oxidation with other enzymes, such as pyranose 2-oxidase.

ANSWER:

Following the reviewer's suggestion, we added the following discussion in the ms (p. 12 in the version with changes highlighted): *"Despite the significant overall sequence identity between G3Oxs and P2Oxs, the topology in the active site surrounding regions differ substantially, which point to distinct mechanisms of controlling regiospecificity. In P2Ox's (e.g., TmP2Ox), C2/C3 regioselectivity is reportedly controlled by the rearrangement of the substrate loop after hydrogen bonds are established between D-Glc and residues D452 and Y456⁴⁰. In PsG3Ox, these latter residues are replaced by the hydrophobic residues A346 and P350, and no hydrogen bond interactions between D-Glc and the loop were detected in the crystallographic, MDs, or docking experiments. However, as mentioned above, docking of D-Glc to structures harboring different*

loop conformations suggest that a semi-open conformation favors the binding of C2 over C3 and at shorter distances to FAD^{N5} and H440 (**Supplementary Figures 27 and 28**), though it is not clear at the moment which type of loop-glucose specific interactions are occurring. Furthermore, the short D-Glc^{H2}-FAD^{N5} distances sampled along the unrestricted MD simulations of the PsG3Ox:FAD: Glc complex are also consistent with a preferred C2 oxidation mechanism starting with D-Glc^{H2} hydride transfer (**Supplementary Figure 30 a,b**).” Thank you.

Reviewer #2

The work elucidates the crystal structure of the novel bacterial Glycoside 3-Oxidases and provides experimental insight into substrate binding with the help of mutagenesis and some MD studies.

The study is novel and could be published in Nature Commun. however some points of revision should be addressed:

i) Regarding the MD simulations - I can not see the all relevant RMSD plots, RMSF plots and Radius of Gyration plots. The current plan MD trajectory is short ideally should be 1 microsecond, nonetheless that accelerated MD is later performed.

ANSWER:

RMSD and RMSF plots were available in the original version of the manuscript (**Supplementary Figs. 16 and 17**, now **18 and 19** for conventional MDs and **Supplementary Figs. 19 and 20**, now **22 and 23** for GaMDs). Our MDs aimed to explore the loop conformations, which are difficult to assess with conventional MDs (cMDs). Thus, as the reviewer observed, we switched to accelerated MDs (in our case GaMDs): with GaMDs, rare events like loop transitions can be observed more quickly than by running longer cMDs. In this regard, in GaMD simulation III*, we have successfully observed a loop transition.

Following the reviewer’s request, we added the corresponding gyration plot radius (**new Supplementary Fig. 20c and 24c**). We have also included the analysis of correlated motions (through dynamic cross-correlation maps (DCCM); see answer iii) for all the simulations (**new Supplementary Figs. 21 and 25**). This data is now in the ms (version with changes highlighted) in **p. 9-10**: “The A352-FAD^{N5} and distances G84-FAD^{N5} distances were used to discriminate between closed ($\leq 20 \text{ \AA}$) and open ($\geq 25 \text{ \AA}$) conformations of the loops, as well as the enzyme’s radius of gyration as smaller values were measured for the closed (and semi-open) states when compared to the open system. Moreover, dynamic cross-correlation analysis (DCCA) of the MD trajectories was carried out as it is a powerful tool to describe and identify functional correlated motions and non-active-site residues that potentially affect catalysis^{42,43}.”

ii) the binding free energies should be calculated at least with MMPBSA/GBSA and correlated to the experiment

ANSWER:

As requested by the reviewer, the binding free energies were calculated by using both the Molecular Mechanics/Poisson–Boltzmann Surface Area (MM-PBSA) and the Molecular Mechanics/Generalised Born Surface Area (MM-GBSA) methods and the results compared to

experimental data. Such calculations are usually performed on multiple short MD simulations of the protein:ligand complex to have a wide sampling of the complex structures. Thus, starting in the docking structures depicted in Fig. 5c,e, we ran 1 μ s of cMD per substrate but divided it into ten replicates of 100 ns each. These trajectories were used to perform the MM-PBSA and MM-GBSA calculations with the AMBER module MMPBSA.py. These calculations are now in the Methods section (p. 28), and the results in the new Supplementary Table 10, new Supplementary Figure 30, and on p. 13: “*In agreement with the experimental trend, the calculation of the differential binding free energy between Glc and Mang using MM-PBSA and MM-GBSA methods, provided an estimation of -3.6 and -4.3 kcal/mol, respectively, in favor of Mang binding (Supplementary Table 10)*”. The results of these trajectories were also used to get more clues about the catalytic mechanism (see the answer to point iv).

iii) the MD and Gaussian-accelerated MD analysis should have been more comprehensive. The authors should explore and analyze in a context the correlated motions of the different system. Some current references can also be used as a guidance and cited on such analysis e.g. <https://chemistry-europe.onlinelibrary.wiley.com/doi/full/10.1002/cphc.202200649> <https://pubs.acs.org/doi/10.1021/jacsau.2c00345>

ANSWER:

The focus of our MD analysis targeted the loop conformations adopted in the different systems, and the reviewer’s suggestion complements this scope well; we have analyzed all MDs (cMDs and GAMDs) in the context of the correlated motions of the different systems and the two suggested references have been added to the main text, and are the new references 42 and 43, p. 10).

Dynamic cross-correlation analysis (DCCA) was performed, and the dynamic cross-correlation maps were plotted for all systems (new Supplementary Figs. 21 and 25). These maps allow identifying the parts of the systems that move in correlated (red) or anti-correlated (blue) manners. For the closed and semi-open systems, most correlation coefficients are small, and no relevant correlated motions were identified. Interestingly, larger values are observed in the more flexible open-state systems. The analysis of the III* trajectory, in which the open-to-close transition is observed after D-Glc removal, reveals the concerted movement of the N-terminal region (starting at the substrate loop), correlated with residues 260-305, and anti-correlated with the insertion-1 segment. This agrees with the loop movements depicted in Fig. 4i, in which the insertion-1 segment and substrate loop close towards each other without substrate. These results were introduced in the text in p. 9 to 11 (marked in yellow).

iv) also ideally the reaction mechanism should have been explored using QM/MM based on the MD snapshots. For a publication in Nature Commun. this should be expected as well. At least authors should comment on this and why it is not done.

ANSWER:

The reviewer’s observation is pertinent, and we ideally would like to comply with their suggestion; however, the additional information taken from QM/MM calculations to the current knowledge does not justify, at our eyes, at present, the (high) time that these experiments will take. Please note that we gathered relevant information on the catalytic mechanism by combining different experimental and computational approaches.

In this version of the ms, we have extended the mechanistic analysis combining our new MD simulations (see answer iii) with DFT quantum mechanical calculations and transient kinetic experiments from reference 32 and proposed a sugar oxidation mechanism as requested by reviewer #4, point 4. On p. 12, the text now reads: “We propose a mechanism for PsG3Ox (Fig. 5a) based on the one described for sugar oxidation by P2Oxs based in density functional theory (DFT) quantum mechanical calculations and transient kinetics³². A hydride transfer (C–H bond breakage) from the C3 position of glycoside (C2 in the case of Glc) to the FAD^{N5} occurs firstly and produces a protonated ketone intermediate (C2=O⁺H) and the reduced flavin; this is followed by proton abstraction promoted by H440, with support of N484, that leads to the keto-sugar product.”

Moreover, new Supplementary Figure 30 is mentioned in the section “Enzyme-substrate interactions” and the following sentences on p. 13:

“The short D-Glc^{H2}–FAD^{N5} distances sampled along the unrestricted MD simulations of the PsG3Ox:FAD:Glc complex are also consistent with a preferred C2 oxidation mechanism starting with D-Glc^{H2} hydride transfer (Supplementary Figure 30a,b).”

“The Mang^{H3}–FAD^{N5} distances sampled along the MD simulations of the PsG3Ox:FAD:Mang complex show values catalytically competent for a mechanism initiated by H3 hydride transfer (Supplementary Figure 30c,d).”

Thank you.

Reviewer #3

Taborda et al have identified a new glycoside 3-oxidase involved in C-glycoside metabolism and conducted biochemical and structural biology studies on it. The biochemical characterization results provide scientifically robust insights into the function and properties of PsG3Ox. However, additional data is required to support the crystallography experiment results, particularly with regard to substrate binding. Overall, the study provides substantial information regarding the function and mechanism of action of glycoside 3-oxidase. However, additional data is necessary to support the experimental results.

1. Validation report suggests that the occupancy of the substrates in the complex structure is not high.

1) For instance, the electron density for D-glucose could not be confirmed in the PDB validation report, and relevant electron density maps should be added to the supporting data.

ANSWER:

The glucose was refined with a substoichiometric occupancy (~ 0.8) to have similar atomic displacement parameters (*a.d.p.s*) with the neighboring atoms, 57-63 Å² vs. 56-70 Å². As requested, the electron density maps of the glucose molecule are provided in the new Supplementary Figure 8.

2) Additionally, there was no negative mFo-Fc observed for mangiferin in the PDB validation

report, and a discussion on this is needed. An electron density map for mangiferin should also be added to the supporting data.

ANSWER:

Mangiferin was refined with a substoichiometric occupancy (~0.8) to have similar *a.d.p.s* with the neighboring atoms, 85-117 Å² vs. 67-95 Å². The following sentence was introduced in the ms (version with changes highlighted) in **p. 5**: “*The D-Glc and Mang were refined with substoichiometric occupancies (~0.8) (Supplementary Figure 8) to have similar a.d.p.s with the neighboring atoms, 57-63 Å² vs. 56-70 Å² and 85-117 Å² vs. 67-95 Å², respectively. The weak electron density in the aglycone of Mang suggests that this region is more flexible, presenting higher a.d.p.s values than the glucose moiety, 108-117 Å² vs 85-96 Å². The lower number of interactions established between the aglycone moiety and the enzyme, Mang^{OH}-K55^{NZ}, could have contributed to the higher flexibility of this region.*”

2. Electron density maps for amino acid regions 60-93 and 346-354, which show large conformation changes depending on the substrate, should be included in the supporting data.

ANSWER:

The electron density maps (contoured at 1.0 RMSD) of the insertion region (60-93) and substrate loop (346-354) for the PsG3Ox, PsG3Ox-Glc, and PsG3Ox-Mang complexes are in the new Supplementary Figure 16, as requested.

3. The quality of the sequence alignment figure needs to be improved as well.

ANSWER:

The quality of the alignment in Fig. 2 was improved.

4. The authors also need to describe the oligomeric state of PsGO3x obtained experimentally and its crystal structure, as they emphasized its importance.

ANSWER:

PsG3Ox is a monomer in solution as reported by Mendes et al. 2016. This finding is consistent with the monomeric subunit observed in the crystal structures. To ensure that this information is conveyed clearly to the readers, the manuscript reads as follow (**p. 6**): “*PsG3Ox shows, when compared to the fungal enzymes (Fig. 2b,c), i) a significantly more solvent-exposed FAD cavity, ii) smaller monomers’ size (~ 500 residues instead of the ~ 600 residues), iii) a monomeric structure (following the reported monomeric oligomerization state in solution²⁰) instead of the tetrameric fungal state, and, iv) significantly higher <a.d.p.>’s values (52 Å²) than fungal enzymes (12-30 Å²).*”

5. Finally, the methods section is missing references to crystallography programs such as MolProbity and Modeller, and the reference section lacks page information for several papers.

ANSWER:

The missing references were added; the page information was added to all papers in the References section. Thank you.

Reviewer #4

In this manuscript, Taborda et al present the kinetic and structural characterization of a flavin-dependent oxidoreductase that performs an alcohol to ketone oxidation reactions for sugars appended to flavonoids, among other molecules. The reviewer has the following comments for the authors to consider:

1. Brevity and clarity - considering the structures for founding members of this enzymes class are already available, the structure description can be significantly shortened and streamlined. Same for the kinetic description. There is no question that the activity of the enzyme is lower for monosaccharides alone, this can be brought out in a much more simply.

ANSWER:

The structure description was shortened on **p. 5 and 6** (highlighted in cyan).

2. Is there a rationale for attempting to "trigger a covalent attachment of FAD" to the enzyme? The motivation for performing this experiment is not clear, and as above, the description of FAD binding can be shortened.

ANSWER:

One notable difference among the members of the POx family is the type of attachment of the FAD cofactor: it is covalently bound in P2Oxs, and in G3Oxs, it is non-covalently attached. Flavin reactivity depends on the cofactor attachment type; e.g., covalent linkage frequently reportedly increases the potential redox of the enzymes. In an attempt to clarify this point, the text now reads (**p. 8**): *"Flavin reactivity depends on the type of cofactor attachment³³ which is a key difference between P2Oxs and G3Oxs (**Fig. 2e**). To investigate whether the covalent attachment of FAD can contribute to increased reactivity of PsG3Ox, i) ¹²⁵AAHT¹²⁸ was replaced by STHW (**Fig 2f**) and, ii) N120 was replaced by a Val, to remove a hydrogen bond to H127 that may preclude its covalent binding to FAD (**Fig. 2e**). However, variants showed non-covalently bound FAD or impaired FAD loading (**Supplementary Figure 14 and Supplementary Table 6**)."*

3. If molecular oxygen is the terminal electron acceptor, does the structure inform the mode/site of oxygen binding? How does oxygen access the reduced flavin?

ANSWER:

We thank the reviewer for the suggestion that led us to analyze more deeply the structure of PsG3Ox. The analysis of the accessible surface area of the PsG3Ox crystal structure identified two main pathways that connect the solvent media with the FAD^{N5}, one cavity that allows access to the bulky substrate and one tunnel that, due to its highly hydrophobic character, could be used for molecular oxygen entrance. After the oxygen reduction, the hydrogen peroxide could exit through an identified polar cavity connecting the FAD with the solvent. This information is now in **Supplementary Table 5**, **new Fig. 2e,f**, **new Supplementary Figure 9**, and in the manuscript in **p. 6**: *"Furthermore, the analysis of the PsG3Ox active site pocket unveiled the presence of a 25 Å length tunnel that connects the enzyme surface and the reactive isoalloxazine moiety of FAD (**Fig. 2e**). This tunnel is mostly delimited by hydrophobic residues (**Supplementary***

Table 5) and has a radius ranging from 1.7 to 2.0 Å. Hydrophobic tunnels allow an efficient way to deliver molecular oxygen to the enzyme's active site³⁵. In PsG3Ox, this hydrophobic tunnel can provide a route for O₂ (with a van der Waals radius of 1.50-1.55 Å) to reach the FAD^{N5}, where it is reduced to hydrogen peroxide. Due to the polarity of H₂O₂, its release should occur alternatively through the FAD cavity (**Fig. 2f**)."

Furthermore, to validate the presence of this hydrophobic tunnel in the PsG3Ox crystal structure, we have performed krypton high pressurization experiments at ESRF (Grenoble, France). One krypton atom was located in an internal pocket connected to the bottom section of the tunnel, thus suggesting that the Kr reached the buried pocket using the hydrophobic tunnel and that this tunnel may function as a diffusion path for the oxygen to reach the FAD. Details were added to the manuscript on **p. 6/7**: "Noble gases like krypton or xenon have been used in the investigation of hydrophobic tunnels due to their hydrophobicity and high atomic numbers³⁶⁻³⁸. The similar dimensions of krypton atoms, with minimal O₂ dimensions, make this a convenient gas to locate possible O₂ tunnels. Pressurization experiments with Kr inserted four krypton atoms in the PsG3Ox crystal structure; however, only one of the Kr atoms (surrounded by hydrophobic residues V299, V349, L351, G366, and F368) is located in an internal pocket connected to the bottom section of the tunnel (**Supplementary Figure 9**). This pocket is also present in the PsG3Ox non-pressurized structure. These results suggest that the hydrophobic tunnel connecting the FAD with the solvent can be used as a diffusion path for molecular oxygen in PsG3Ox."

4. Together with Figure 5 where the binding of the substrates relative to the flavin cofactor is illustrated, it would be helpful to see a rendering of the proposed reaction mechanism with the hydride transfer step formally illustrated. It would then be useful to discuss the substrate binding modes relative to the flavin-N5.

ANSWER:

As suggested by the reviewer, a new panel (a) was added to **Fig 5** to show the proposed reaction mechanism of mangiferin oxidation.

5. Minor comment - some of the axis labels in Figure 4 are too small to be legible.

ANSWER:

This is corrected. Thank you.

Reviewers' Comments:

Reviewer #2:

Remarks to the Author:

The authors responded to my comments and made relevant revision (including new analysis)of the manuscript. I therefore recommend publication.

Reviewer #4:

Remarks to the Author:

All reviewer concerns have been addressed.

REVIEWERS' COMMENTS

Reviewer #2 (Remarks to the Author):

The authors responded to my comments and made relevant revision (including new analysis) of the manuscript. I therefore recommend publication.

Response: We thank the reviewer for the comments and suggestions that resulted in the improved quality of the manuscript.

Reviewer #4 (Remarks to the Author):

All reviewer concerns have been addressed.

Response: We thank the reviewer for the comments and suggestions that resulted in improved quality of the manuscript.